# Switching Autoregressive Low-rank Tensor Models

**Hyun Dong Lee**
Computer Science Department
Stanford University
hdlee@stanford.edu

**Andrew Warrington**
Department of Statistics
Stanford University
awarring@stanford.edu

**Joshua I. Glaser**
Department of Neurology
Northwestern University
j-glaser@northwestern.edu

**Scott W. Linderman**
Department of Statistics
Stanford University
scott.linderman@stanford.edu

## Abstract

An important problem in time-series analysis is modeling systems with time-varying dynamics. Probabilistic models with joint continuous and discrete latent states offer interpretable, efficient, and experimentally useful descriptions of such data. Commonly used models include autoregressive hidden Markov models (ARHMMs) and switching linear dynamical systems (SLDSs), each with its own advantages and disadvantages. ARHMMs permit exact inference and easy parameter estimation, but are parameter intensive when modeling long dependencies, and hence are prone to overfitting. In contrast, SLDSs can capture long-range dependencies in a parameter efficient way through Markovian latent dynamics, but present an intractable likelihood and a challenging parameter estimation task. In this paper, we propose *switching autoregressive low-rank tensor* (SALT) models, which retain the advantages of both approaches while ameliorating the weaknesses. SALT parameterizes the tensor of an ARHMM with a low-rank factorization to control the number of parameters and allow longer range dependencies without overfitting. We prove theoretical and discuss practical connections between SALT, linear dynamical systems, and SLDSs. We empirically demonstrate quantitative advantages of SALT models on a range of simulated and real prediction tasks, including behavioral and neural datasets. Furthermore, the learned low-rank tensor provides novel insights into temporal dependencies within each discrete state.

## 1 Introduction

Many time series analysis problems involve jointly segmenting data and modeling the time-evolution of the system within each segment. For example, a common task in computational ethology [Datta et al., 2019] — the study of natural behavior — is segmenting videos of freely moving animals into states that represent distinct behaviors, while also quantifying the differences in dynamics between states [Wiltschko et al., 2015, Costacurta et al., 2022]. Similarly, discrete shifts in the dynamics of neural activity may reflect changes in underlying brain state [Saravani et al., 2019, Recanatesi et al., 2022]. Model-based segmentations are experimentally valuable, providing an unsupervised grouping of neural or behavioral states together with a model of the dynamics within each state.

One common probabilistic state space model for such analyses is the *autoregressive hidden Markov model* (ARHMM) [Ephraim et al., 1989]. For example, MoSeq [Wiltschko et al., 2015] uses ARHMMs for unsupervised behavioral analysis of freely moving animals. ARHMMs learn a set of linear autoregressive models, indexed by a discrete state, to predict the next observation as a function of previous observations. Inference in ARHMMs then reduces to inferring which AR

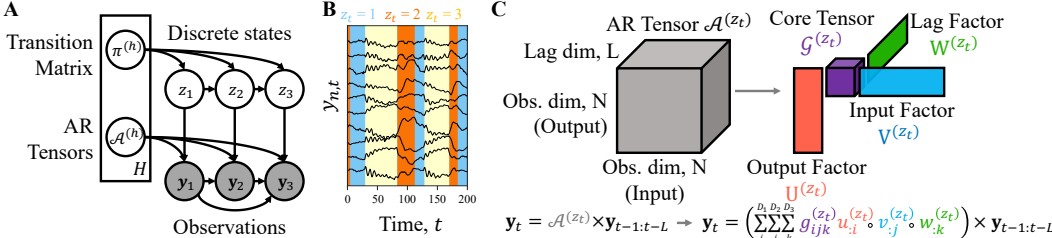

*Figure 1:* **SALT imposes a low-rank constraint on the autoregressive tensor**: **(A)** The probabilistic graphical model of an ARHMM. **(B)** An example multi-dimensional time series generated from an ARHMM. Background color indicates which discrete state (and hence autoregressive tensor) was selected at each time. **(C)** In SALT, each autoregressive dynamics tensor of an ARHMM is parameterized as a low-rank tensor.

process best explains the observed data at each timestep (in turn also providing the segmentation). The simplicity of ARHMMs allows for exact state inference via message passing, and closed-form updates for parameter estimation using expectation-maximization (EM). However, the ARHMM requires high order autoregressive dependencies to model long timescale dependencies, and its parameter complexity is quadratic in the data dimension, making it prone to overfitting.

*Switching linear dynamical systems* (SLDS) [Ghahramani and Hinton, 2000] ameliorate some of the drawbacks of the ARHMM by introducing a low-dimensional, continuous latent state. These models have been used widely throughout neuroscience [Saravani et al., 2019, Petreska et al., 2011, Linderman et al., 2019, Glaser et al., 2020, Nair et al., 2023]. Unlike the ARHMM, the SLDS can capture long timescale dependencies through the dynamics of the continuous latent state, while also being much more parameter efficient than ARHMMs. However, exact inference in SLDSs is intractable due to the exponential number of potential discrete state paths governing the time-evolution of the continuous latent variable. This intractability has led to many elaborate and specialized approximate inference techniques [Ghahramani and Hinton, 2000, Barber, 2006, Fox, 2009, Murphy and Russell, 2001, Linderman et al., 2017, Zoltowski et al., 2020]. Thus, the SLDS gains parameter efficiency at the expense of the computational tractability and statistical simplicity of the ARHMM.

We propose a new class of unsupervised probabilistic models that we call *switching autoregressive low-rank tensor* (SALT) models. Our novel insight is that when you marginalize over the latent states of a linear dynamical system, you obtain an autoregressive model with full history dependence. However, these autoregressive dependencies are not arbitrarily complex — they factor into a low-rank tensor that can be well-approximated with a finite-history model. We formalize this connection in Proposition 1. SALT models are constrained ARHMMs that leverage this insight. Rather than allowing for arbitrary autoregressive dependencies, SALT models are constrained to be low-rank. The low-rank property allows us to construct a low-dimensional continuous description of the data, jointly with the discrete segmentation provided by the switching states. Thus, SALT models inherit the experimentally useful representations and parsimonious parameter complexity of an SLDS, as well as the ease of inference and estimation of ARHMMs. We demonstrate the advantages of SALT models empirically using synthetic data as well as real neural and behavioral time series. Finally, in addition to improving predictive performance, we show how the low-rank nature of SALT models can offer new insights into complex systems, like biological neural networks.

## 2   Background

This section introduces the notation used throughout the paper and describes preliminaries on low-rank tensor decomposition, vector autoregressive models, switching autoregressive models, linear dynamical systems, and switching linear dynamical systems.

**Notation**   We follow the notation of Kolda and Bader [2009]. We use lowercase letters for scalar variables (e.g. $a$), uppercase letters for scalar constants (e.g. $A$), boldface lowercase letters for vectors (e.g. $\mathbf{a}$), boldface uppercase letters for matrices (e.g. $\mathbf{A}$), and boldface Euler script for tensors of

---

Source code is available at `https://github.com/lindermanlab/salt`.

order three or higher (e.g. $\mathcal{A}$). We use $\mathbf{A}_{i::}$, $\mathbf{A}_{:j:}$, and $\mathbf{A}_{::k}$ to denote the horizontal, lateral, and frontal slices respectively of a three-way tensor $\mathcal{A}$. Similarly, we use $\mathbf{a}_{i:}$ and $\mathbf{a}_{:j}$ to denote the $i^{th}$ row and $j^{th}$ column of a matrix $\mathbf{A}$. $\mathbf{a} \circ \mathbf{b}$ represents the vector outer product between vectors $\mathbf{a}$ and $\mathbf{b}$. The $n$-mode tensor-matrix (tensor-vector) product is represented as $\mathcal{A} \times_n \mathbf{A}$ ($\mathcal{A} \bar{\times}_n \mathbf{a}$). We denote the vectorization of an $n$-way tensor $\mathcal{G}$, with dimensions $D_{1:n}$, as $\text{vec}(\mathcal{G})$. This is performed by successively flattening the last dimensions of the tensor, and results in a vector of size equal to the product of the dimensions of the tensor. We denote the mode-$n$ matricization of a tensor $\mathcal{G}$ as $\mathcal{G}_{(n)}$. This is defined as the stack of vectors resulting from vectorizing the matrix (or tensor) defined by each slice through the $n^{\text{th}}$ dimension. This results in a matrix with leading dimension $D_n$, and second dimension equal to the product of the sizes of the other dimensions. We will denote a $T$-length time series of $N$-dimensional observed data as $\mathbf{Y} \in \mathbb{R}^{N \times T}$. Note that we will use the shorthand $\mathbf{y}_t \in \mathbb{R}^N$ to denote the observation at time $t$, and $y_{j,t} \in \mathbb{R}$ to denote the $j^{\text{th}}$ element in the $t^{\text{th}}$ observation. It will be clear from context which dimension is being indexed.

**Tensor Decomposition**    For $\mathcal{A} \in \mathbb{R}^{N_1 \times N_2 \times N_3}$, the Tucker decomposition is defined as,

$$\mathcal{A} = \sum_{i=1}^{D_1} \sum_{j=1}^{D_2} \sum_{k=1}^{D_3} g_{ijk} \, \mathbf{u}_{:i} \circ \mathbf{v}_{:j} \circ \mathbf{w}_{:k}, \tag{1}$$

where $\mathbf{u}_{:i}$, $\mathbf{v}_{:j}$, and $\mathbf{w}_{:k}$ are the columns of the factor matrices $\mathbf{U} \in \mathbb{R}^{N_1 \times D_1}$, $\mathbf{V} \in \mathbb{R}^{N_2 \times D_2}$, and $\mathbf{W} \in \mathbb{R}^{N_3 \times D_3}$, respectively, and $g_{ijk}$ are the entries in the core tensor $\mathcal{G} \in \mathbb{R}^{D_1 \times D_2 \times D_3}$.

The CANDECOMP/PARAFAC (CP) decomposition is a special case of the Tucker decomposition, with $D_1 = D_2 = D_3$ and a diagonal core tensor $\mathcal{G}$.

**Vector autoregressive models**    Let $\mathbf{Y} \in \mathbb{R}^{N \times T}$ denote a multivariate time series with $\mathbf{y}_t \in \mathbb{R}^N$ for all $t$. An order-$L$ vector autoregressive (VAR) model with Gaussian innovations is defined by,

$$\mathbf{y}_t \sim \mathcal{N} \left( \sum_{j=1}^{N} \sum_{k=1}^{L} \mathbf{a}_{:jk} y_{j,t-k} + \mathbf{b}, \, \mathbf{R} \right), \tag{2}$$

where $\mathcal{A} \in \mathbb{R}^{N \times N \times L}$ is the autoregressive tensor, whose frontal slice $\mathbf{A}_{::l}$ is the dynamics matrix for lag $l$, $\mathbf{b} \in \mathbb{R}^N$ is the bias, and $\mathbf{R} \in \mathbb{R}^{N \times N}_{\succeq 0}$ is a positive semi-definite covariance matrix. The parameters $\boldsymbol{\Theta} = (\mathcal{A}, \mathbf{b}, \mathbf{R})$ can be estimated via ordinary least squares [Hamilton, 2020].

We note that, to our knowledge, there is no clear consensus on the best way to regularize the potentially large parameter space of vector autoregressive (hidden Markov) models; several possibilities exist, see, e.g., Melnyk and Banerjee [2016] or Ni and Sun [2005]. Many regularizers and priors are difficult to work with, and so are not widely used in practice. Beyond this, even well-regularized ARHMMs do not natively capture interpretable low-dimensional dynamics, as both SALT and SLDS models do (see Figure 3). These low-dimensional continuous representations are as experimentally useful as the discrete segmentation, and hence are a key desiderata for any method we consider.

**Switching autoregressive models**    One limitation of VAR models is that they assume the time series is stationary; i.e. that one set of parameters holds for all time steps. Time-varying autoregressive models allow the autoregressive process to change at various time points. One such VAR model, referred to as a switching autoregressive model or autoregressive hidden Markov model (ARHMM), switches the parameters over time according to a discrete latent state [Ephraim et al., 1989]. Let $z_t \in \{1, \ldots, H\}$ denote the discrete state at time $t$, an ARHMM defines the following generative model,

$$z_t \sim \text{Cat}\left( \boldsymbol{\pi}^{(z_{t-1})} \right), \qquad \mathbf{y}_t \sim \mathcal{N} \left( \sum_{j=1}^{N} \sum_{k=1}^{L} \mathbf{a}_{:jk}^{(z_t)} y_{j,t-k} + \mathbf{b}^{(z_t)}, \, \mathbf{R}^{(z_t)} \right), \tag{3}$$

where $\boldsymbol{\pi}^{(h)} \in \{\boldsymbol{\pi}^{(h)}\}_{h=1}^{H}$ is the the $h$-th row of the discrete state transition matrix.

A switching VAR model is simply a type of hidden Markov model, and as such it is easily fit via the expectation-maximization (EM) algorithm within the Baum-Welch algorithm. The M-step amounts to solving a weighted least squares problem.

**Linear dynamical systems**   The number of parameters in a VAR model grows as $\mathcal{O}(N^2 L)$. For high-dimensional time series, this can quickly become intractable. Linear dynamical systems (LDS) [Murphy, 2012] offer an alternative means of modeling time series via a continuous latent state $\mathbf{x}_t \in \mathbb{R}^S$,

$$\mathbf{x}_t \sim \mathcal{N}(\mathbf{A}\mathbf{x}_{t-1} + \mathbf{b}, \mathbf{Q}), \qquad \mathbf{y}_t \sim \mathcal{N}(\mathbf{C}\mathbf{x}_t + \mathbf{d}, \mathbf{R}), \tag{4}$$

where $\mathbf{Q} \in \mathbb{R}_{\succeq 0}^{S \times S}$ and $\mathbf{R} \in \mathbb{R}_{\succeq 0}^{N \times N}$. Here, the latent states follow a first-order VAR model, and the observations are conditionally independent given the latent states. As we discuss in Section 3.3, marginalizing over the continuous latent states renders $\mathbf{y}_t$ dependent on the preceding observations, just like in a high order VAR model.

Compared to the VAR model, however, the LDS has only $\mathcal{O}(S^2 + NS + N^2)$ parameters if $\mathbf{R}$ is a full covariance matrix. This further reduces to $\mathcal{O}(S^2 + NS)$ if $\mathbf{R}$ is diagonal. As a result, when $S \ll N$, the LDS has many fewer parameters than a VAR model. Thanks to the linear and Gaussian assumptions of the model, the parameters can be easily estimated via EM, using the Kalman smoother to compute the expected values of the latent states.

**Switching linear dynamical systems**   A switching LDS combines the advantages of the low-dimensional continuous latent states of an LDS, with the advantages of discrete switching from an ARHMM. Let $z_t \in \{1, \ldots, H\}$ be a discrete latent state with Markovian dynamics (3), and let it determine some or all of the parameters of the LDS (e.g. $\mathbf{A}$ would become $\mathbf{A}^{(z_t)}$ in (4)). We note that SLDSs often use a *single-subspace*, where $\mathbf{C}$, $\mathbf{d}$ and $\mathbf{R}$ are shared across states, reducing parameter complexity and simplifying the optimization.

Unfortunately, parameter estimation is considerably harder in SLDS models. The posterior distribution over all latent states, $p(\mathbf{z}_{1:T}, \mathbf{x}_{1:T} \mid \mathbf{y}_{1:T}, \mathbf{\Theta})$, where $\mathbf{\Theta}$ denotes the parameters, is intractable [Lerner, 2003]. Instead, these models are fit via approximate inference methods like MCMC [Fox, 2009, Linderman et al., 2017], variational EM [Ghahramani and Hinton, 2000, Zoltowski et al., 2020], particle EM [Murphy and Russell, 2001, Doucet et al., 2001], or other approximations [Barber, 2006]. Selecting the appropriate fitting and inference methodologies is itself non-trivial hyperparameter. Furthermore, each method also brings additional estimation hyperparameters that need to be tuned prior to even fitting the generative model. We look to define a model that enjoys the benefits of SLDSs, but avoids the inference and estimation difficulties.

## 3   SALT: Switching Autoregressive Low-rank Tensor Models

Here we formally introduce SALT models. We begin by defining the generative model (also illustrated in Figure 1), and describing how inference and model fitting are performed. We conclude by drawing connections between SALT and SLDS models.

### 3.1   Generative Model

SALT factorizes each autoregressive tensor $\boldsymbol{\mathcal{A}}^{(h)}$ for $h \in \{1, \ldots, H\}$ of an ARHMM as a product of low-rank factors. Given the current discrete state $z_t$, each observation $\mathbf{y}_t \in \mathbb{R}^N$ is modeled as being normally distributed conditioned on $L$ previous observations $\mathbf{y}_{t-1:t-L}$,

$$z_t \sim \text{Cat}\left(\boldsymbol{\pi}^{(z_{t-1})}\right), \tag{5}$$

$$\mathbf{y}_t \overset{\text{i.i.d.}}{\sim} \mathcal{N}\left(\sum_{j=1}^{N}\sum_{k=1}^{L} \mathbf{a}_{\text{SALT},:jk}^{(z_t)} y_{j,t-k} + \mathbf{b}^{(z_t)}, \mathbf{\Sigma}^{(z_t)}\right), \tag{6}$$

$$\boldsymbol{\mathcal{A}}_{\text{SALT}}^{(z_t)} = \sum_{i=1}^{D_1}\sum_{j=1}^{D_2}\sum_{k=1}^{D_3} g_{ijk}^{(z_t)} \mathbf{u}_{:i}^{(z_t)} \circ \mathbf{v}_{:j}^{(z_t)} \circ \mathbf{w}_{:k}^{(z_t)}, \tag{7}$$

where $\mathbf{u}_{:i}^{(z_t)}$, $\mathbf{v}_{:j}^{(z_t)}$, and $\mathbf{w}_{:k}^{(z_t)}$ are the columns of the factor matrices $\mathbf{U}^{(z_t)} \in \mathbb{R}^{N \times D_1}$, $\mathbf{V}^{(z_t)} \in \mathbb{R}^{N \times D_2}$, and $\mathbf{W}^{(z_t)} \in \mathbb{R}^{L \times D_3}$, respectively, and $g_{ijk}^{(z_t)}$ are the entries in the core tensor $\boldsymbol{\mathcal{G}}^{(z_t)} \in \mathbb{R}^{D_1 \times D_2 \times D_3}$. The vector $\boldsymbol{b}^{(z_t)} \in \mathbb{R}^N$ and positive definite matrix $\mathbf{\Sigma}^{(z_t)} \in \mathbb{R}_{\succeq 0}^{N \times N}$ are the bias and covariance for state $z_t$. Without further restriction this decomposition is a Tucker decomposition [Kolda and Bader, 2009]. If $D_1 = D_2 = D_3$ and $\boldsymbol{\mathcal{G}}_{z_t}$ is diagonal, it corresponds to a

*Table 1:* Comparison of number of parameters for the methods we consider. We exclude covariance matrix parameters, as the parameterization of the covariance matrix is independent of method. Throughout our experiments, we find $S \approx D$.

| Model | Parameter Complexity | (Example from Section 5.4) |
|---|---|---|
| SLDS | $\mathcal{O}(NS + HS^2)$ | 2.8K |
| CP-SALT | $\mathcal{O}(H(ND + LD))$ | 8.1K |
| Tucker-SALT | $\mathcal{O}(H(ND + LD + D^3))$ | 17.4K |
| Order-$L$ ARHMM | $\mathcal{O}(HN^2L)$ | 145.2K |

CP decomposition [Kolda and Bader, 2009]. We refer to ARHMM models with these factorizations as Tucker-SALT and CP-SALT respectively. Note that herein we will only consider models where $D_1 = D_2 = D_3 = D$, where we refer to $D$ as the "rank" of the SALT model (for both Tucker-SALT and CP-SALT). In practice, we find that models constrained in this way perform well, and so this constraint is imposed simply to reduce the search space of models and could easily be relaxed.

Table 1 shows the number of parameters for order-$L$ ARHMMs, SLDSs, and SALT. Focusing on the lag dependence, the number of ARHMM parameters grows as $\mathcal{O}(HN^2L)$, whereas SALT grows as only $\mathcal{O}(HDL)$ with $D \ll N$. SALT can also make a simplifying single-subspace constraint, where certain emission parameters are shared across discrete states.

**Low-dimensional Representation** Note that SALT implicitly defines a low-dimensional continuous representation, analogous to the continuous latent variable in SLDS,

$$\mathcal{P} = \sum_{j=1}^{D_2} \sum_{k=1}^{D_3} \mathbf{g}_{:jk}^{(z_t)} \circ \mathbf{v}_{:j}^{(z_t)} \circ \mathbf{w}_{:k}^{(z_t)}, \tag{8}$$

$$\mathbf{x}_t = \sum_{j=1}^{N} \sum_{k=1}^{L} \mathbf{P}_{:jk} y_{j,t-k}. \tag{9}$$

The low-dimensional $\mathbf{x}_t \in \mathbb{R}^{D_1}$ vectors can be visualized, similar to the latent states in SLDS models, to further interrogate the learned dynamics, as we show in Figure 3. Note the vector $\mathbf{x}_t \in \mathbb{R}^{D_1}$, when multiplied by the output factors $\mathbf{U}^{(z_t)}$, is the mean of the next observation.

### 3.2 Model Fitting and Inference

Since SALT models are ARHMMs, we can apply the expectation-maximization (EM) algorithm to fit model parameters and perform state space inference. We direct the reader to Murphy [2012] for a detailed exposition of EM and include only the key points here.

The E-step solves for the distribution over latent variables given observed data and model parameters. For SALT, this is the distribution over $z_t$, denoted $\omega_t^{(h)} = \mathbb{E}[z_t = h \mid \mathbf{y}_{1:T}, \boldsymbol{\theta}]$. This can be computed exactly with the forward-backward algorithm, which is fast and stable. The marginal likelihood can be evaluated exactly by taking the product across $t$ of expectations of (6) under $\omega_t^{(h)}$.

The M-step then updates the parameters of the model given the distribution over latent states. For SALT, the emission parameters are $\boldsymbol{\theta} = \{\mathbf{U}^{(h)}, \mathbf{V}^{(h)}, \mathbf{W}^{(h)}, \mathcal{G}^{(h)}, \boldsymbol{b}^{(h)}, \boldsymbol{\Sigma}^{(h)}, \boldsymbol{\pi}^{(h)}\}_{h=1}^{H}$. We use closed-form coordinate-wise updates to maximize the expected log likelihood evaluated in the E-step. Each factor update amounts to solving a weighted least squares problem. We include just one update step here for brevity, and provide all updates in full in Appendix A. Assuming here that $\mathbf{b}^{(h)} = \mathbf{0}$ for simplicity, the update rule for the lag factors is as follows:

$$\mathbf{w}^{(h)\star} = \left( \sum_t \omega_t^{(h)} \widetilde{\mathbf{X}}_t^{(h)\top} (\boldsymbol{\Sigma}^{(h)})^{-1} \widetilde{\mathbf{X}}_t^{(h)} \right)^{-1} \left( \sum_t \omega_t^{(h)} \widetilde{\mathbf{X}}_t^{(h)\top} (\boldsymbol{\Sigma}^{(h)})^{-1} \mathbf{y}_t \right) \tag{10}$$

where $\widetilde{\mathbf{X}}_t^{(h)} = \mathbf{U}^{(h)} \mathcal{G}_{(1)}^{(h)} (\mathbf{V}^{(h)\top} \mathbf{y}_{t-1:t-L} \otimes \mathbf{I}_{D_3})$ and $\mathbf{w}^{(h)\star} = \text{vec}(\mathbf{W}^{(h)})$. Crucially, these coordinate wise updates are exact, and so we recover the fast and monotonic convergence of EM.

## 3.3 Connections Between SALT and Linear Dynamical Systems

SALT is not only an intuitive regularization for ARHMMs, it is grounded in a mathematical correspondence between autoregressive models and linear dynamical systems.

**Proposition 1** (Low-Rank Tensor Autoregressions Approximate Stable Linear Dynamical Systems)**.** *Consider a stable linear time-invariant Gaussian dynamical system. We define the steady-state Kalman gain matrix as $\mathbf{K} = \lim_{t \to \infty} \mathbf{K}_t$, and $\mathbf{\Gamma} = \mathbf{A}(\mathbf{I} - \mathbf{KC})$. The matrix $\mathbf{\Gamma} \in \mathbb{R}^{S \times S}$ has eigenvalues $\lambda_1, \ldots, \lambda_S$. Let $\lambda_{\max} = \max_s |\lambda_s|$; for a stable LDS, $\lambda_{\max} < 1$ [Davis and Vinter, 1985]. Let $n$ denote the number of real eigenvalues and $m$ the number of complex conjugate pairs. Let $\hat{\mathbf{y}}_t^{(\mathsf{LDS})} = \mathbb{E}[\mathbf{y}_t \mid \mathbf{y}_{1:t-1}]$ denote the predictive mean under a steady-state LDS, and $\hat{\mathbf{y}}_t^{(\mathsf{SALT})}$ the predictive mean under a SALT model. An order-$L$ Tucker-SALT model with rank $n + 2m = S$, or a CP-SALT model with rank $n + 3m$, can approximate the predictive mean of the steady-state LDS with error $\|\hat{\mathbf{y}}_t^{(\mathsf{LDS})} - \hat{\mathbf{y}}_t^{(\mathsf{SALT})}\|_\infty = \mathcal{O}(\lambda_{\max}^L)$.*

*Proof.* We give a sketch of the proof here and a full proof in Appendix B. The analytic form of $\mathbb{E}\left[\mathbf{y}_t \mid \mathbf{y}_{1:t-1}\right]$ is a linear function of $\mathbf{y}_{t-l}$ for $l = 1, \ldots, \infty$. For this sketch, consider the special case where $\mathbf{b} = \mathbf{d} = \mathbf{0}$. Then the coefficients of the linear function are $\mathbf{C}\mathbf{\Gamma}^l\mathbf{K}$. As all eigenvalues of $\mathbf{\Gamma}$ have magnitude less than one, the coefficients decay exponentially in $l$. We can therefore upper bound the approximation error introduced by truncating the linear function to $L$ terms to $\mathcal{O}(\lambda_{\max}^L)$. To complete the proof, we show that the truncated linear function can be represented exactly by a tensor regression with at most a specific rank. Thus, only truncated terms contribute to the error. $\quad\square$

This proposition shows that the steady-state predictive distribution of a stable LDS can be approximated by a low-rank tensor autoregression, with a rank determined by the eigenspectrum of the LDS. We validate this proposition experimentally in Section 5.1. Note as well that the predictive distribution will converge to a fixed covariance, and hence can also be exactly represented by the covariance matrices $\mathbf{\Sigma}^{(h)}$ estimated in SALT models.

**Connections with Switching Linear Dynamical Systems**   With this foundation, it is natural to hypothesize that a *switching* low-rank tensor autoregression like SALT could approximate a *switching* LDS. There are two ways this intuition could fail: first, if the dynamics in a discrete state of an SLDS are unstable, then Proposition 1 would not hold; second, after a discrete state transition in an SLDS, it may take some time before the dynamics reach stationarity. We empirically test how well SALT approximates an SLDS in Section 5 and find that, across a variety of datasets, SALT obtains commensurate performance with considerably simpler inference and estimation algorithms.

## 4   Related Work

**Low-rank tensor decompositions of time-invariant autoregressive models**   Similar to this work, Wang et al. [2021] also modeled the transition matrices as a third-order tensor $\mathcal{A} \in \mathbb{R}^{N \times N \times L}$ where the $\mathbf{A}_{::l}$ is the $l$-th dynamics matrix. They then constrained the tensor to be low-rank via a Tucker decomposition, as defined in (1). However, unlike SALT, their model was time-invariant, did not have an ARHMM structure, or, make connections to the LDS and SLDS, as in Proposition 1.

**Low-rank tensor decompositions of time-varying autoregressive models**   Low-rank tensor-based approaches have also been used to model time-varying AR processes [Harris et al., 2021, Zhang et al., 2021]. Harris et al. [2021] introduced TVART, which first splits the data into $T$ contiguous fixed-length segments, each with its own AR-1 process. TVART can be thought of as defining a $T \times N \times N$ ARHMM dynamics tensor and progressing through discrete states at fixed time points. This tensor is parameterized using the CP decomposition and optimized using an alternating least squares algorithm, with additional penalties such that the dynamics of adjacent windows are similar. By contrast, SALT automatically segments, rather than windows, the time-series into learned and re-usable discrete states.

Zhang et al. [2021] constructed a Bayesian model of higher-order AR matrices that can vary over time. First, $H$ VAR dynamics tensors are specified, parameterized as third-order tensors with a rank-1 CP decomposition. The dynamics at a given time are then defined as a weighted sum of the tensors, where the weights have a prior density specified by an Ising model. Finally, inference over the weights is performed using MCMC. This method can be interpreted as a factorial ARHMM, hence offering substantial modeling flexibility, but sacrificing computational tractability when $H$ is large.

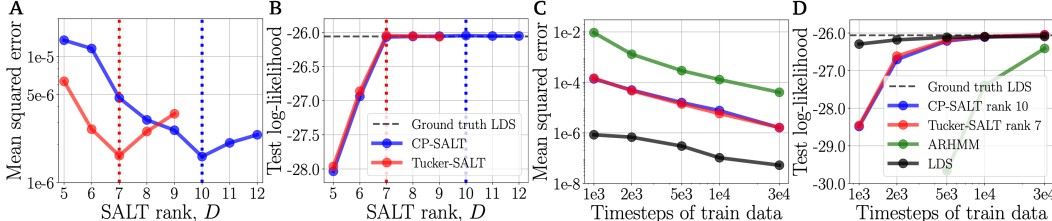

*Figure 2:* **SALT approximates LDS**: Data simulated from an LDS for which $n = 1$ and $m = 3$ (see Proposition 1). **(A-B)**: Average mean squared error of the autoregressive tensor corresponding to the LDS simulation and the log-likelihood of test data, as a function of SALT rank. According to Proposition 1, to model the LDS Tucker-SALT and CP-SALT require 7 and 10 ranks respectively (indicated by vertical dashed lines). Note the *parameter* error increases above the predicted threshold as a result of overfitting. **(C-D)**: Mean squared error of the learned autoregressive tensor and log-likelihood of test data as a function of training data.

**Low-rank tensor decompositions of neural networks**   Low-rank tensor decomposition methods have also been used to make neural networks more parameter efficient. Novikov et al. [2015] used the tensor-train decomposition [Oseledets, 2011] on the dense weight matrices of the fully-connected layers to reduce the number of parameters. Yu et al. [2017] and Qiu et al. [2021] applied the tensor-train decomposition to the weight tensors for polynomial interactions between the hidden states of recurrent neural networks (RNNs) to efficiently capture high-order temporal dependencies. Unlike switching models with linear dynamics, recurrent neural networks have dynamics that are hard to interpret, their state estimates are not probabilistic, and they do not provide experimentally useful data segmentations.

**Linear dynamical systems and low-rank linear recurrent neural networks**   Valente et al. [2022] recently examined the relationship between LDSs and low-rank linear RNNs. They provide the conditions under which low-rank linear RNNs can exactly model the first-order autoregressive distributions of LDSs, and derive the transformation to convert between model classes under those conditions. This result has close parallels to Proposition 1. Under the conditions identified by Valente et al. [2022], the approximation in Proposition 1 becomes exact with just one lag term. However, when those conditions are not satisfied, we show that one still recovers an LDS approximation with a bounded error that decays exponentially in the number of lag terms.

## 5   Results

We now empirically validate SALT by first validating the theoretical claims made in Section 3, and then apply SALT to two synthetic examples to compare SALT to existing methods. We conclude by using SALT to analyze real mouse behavioral recordings and *C. elegans* neural recordings.

### 5.1   SALT Faithfully Approximates LDS

To test the theoretical result that SALT can closely approximate a linear dynamical system, we fit SALT models to data sampled from an LDS. The LDS has $S = 7$ dimensional latent states with random rotational dynamics, where $\Gamma$ has $n = 1$ real eigenvalue and $m = 3$ pairs of complex eigenvalues, and $N = 20$ observations with a random emission matrix.

For Figure 2, we trained CP-SALT and Tucker-SALT with $L = 50$ lags and varying ranks. We first analyzed how well SALT reconstructed the parameters of the autoregressive dynamics tensor. As predicted by Proposition 1, Figure 2A shows that the mean squared errors between the SALT tensor and the autoregressive tensor corresponding to the simulated LDS are the lowest when the ranks of CP-SALT and Tucker-SALT are $n + 3m = 10$ and $n + 2m = 7$ respectively. We then computed log-likelihoods on 5,000 timesteps of held-out test data (Figure 2B). Interestingly, the predictive performance of both CP-SALT and Tucker-SALT reach the likelihood of the ground truth LDS model with rank $n + 2m = 7$, suggesting that sometimes smaller tensors than suggested by Proposition 1 may still be able to provide good approximations to the data. We also show in Figures 2C and 2D that, as predicted, SALT models require much less data to fit than ARHMMs. We show extended empirical results and discussion on Proposition 1 in Appendix D.1.

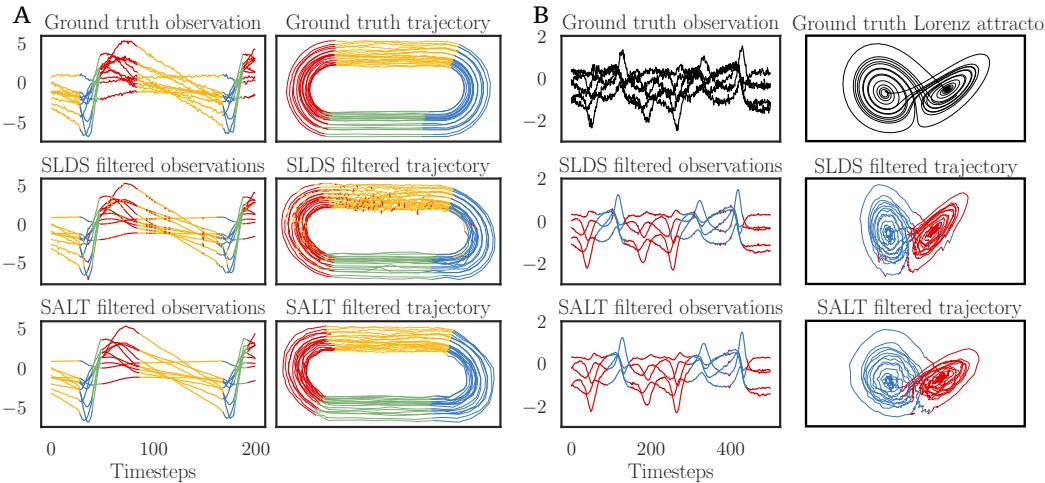

*Figure 3:* **SALT reconstructs simulated SLDS data and Lorenz attractor**: **(Top row)** Observation generated from a low-dimensional trajectory. (A) shows ten observations generated from a recurrent "NASCAR" SLDS trajectory Linderman et al. [2017]. (B) 20-dimensional observations generated from a Lorenz attractor (5 observed dimensions are shown). **(Middle and bottom rows)**: filtered observations and inferred low-dimensional trajectories from SLDS and SALT models. Colors indicate discrete state for ground truth (if available) and fitted models. SLDS and SALT find comparable filtered trajectories and observations. It is important to note that the latent spaces in both SLDS and SALT are only identifiable up to a linear transformation. We therefore align the latent trajectories for ease of comparison. This latent structure is reliably found by both SALT and SLDS.

## 5.2 Synthetic Switching LDS Examples

Proposition 1 quantifies the convergence properties of low-rank tensor regressions when approximating stable LDSs. Next we tested how well SALT can approximate the more expressive *switching* LDSs. We first applied SALT to data generated from a recurrent SLDS [Linderman et al., 2017], where the two-dimensional ground truth latent trajectory resembles a NASCAR® track (Figure 3A). SALT accurately reconstructed the ground truth filtered trajectories and discrete state segmentation, and yielded very similar results to an SLDS model. We also tested the ability of SALT to model nonlinear dynamics – specifically, a Lorenz attractor – which SLDSs are capable of modeling. Again, SALT accurately reconstructed ground truth latents and observations, and closely matched SLDS segmentations. These results suggest that SALT models provide a good alternative to SLDS models. Finally, in Appendix D.3, we used SLDS-generated data to compare SALT and TVART [Harris et al., 2021], another tensor-based method for modeling autoregressive processes, and find that SALT more accurately reconstructed autoregressive dynamics tensors than TVART.

## 5.3 Modeling Mouse Behavior

Next we considered a video segmentation problem commonly faced in the field of computational neuroethology [Datta et al., 2019]. Wiltschko et al. [2015] collected videos of mice freely behaving in a circular open field. They projected the video data onto the top 10 principal components (Figure 4A) and used an ARHMM to segment the PCA time series into distinct behavioral states. Here, we compared ARHMMs and CP-SALT with data from three mice. We used the first 35,949 timesteps of each recording, which were collected at 30Hz resolution. We used $H = 50$ discrete states and fitted ARHMMs and CP-SALT models with varying lags and ranks.

The likelihood on a held-out validation set shows that the ARHMM overfitted quickly as the number of lags increased, while CP-SALT was more robust to overfitting (Figure 4B). We compared log-likelihoods of the best model (evaluated on the validation set) on a separate held-out test set and found that CP-SALT consistently outperformed ARHMM across mice (Figure 4C).

We also investigated the quality of SALT segmentations of the behavioral data (Appendix E.3). We found that the PCA trajectories upon transition into a discrete SALT state were highly stereotyped, suggesting that SALT segments the data into consistent behavioral states. Furthermore, CP-SALT used fewer discrete states than the ARHMM, suggesting that the ARHMM may have oversegmented and that CP-SALT offers a more parsimonious description of the data.

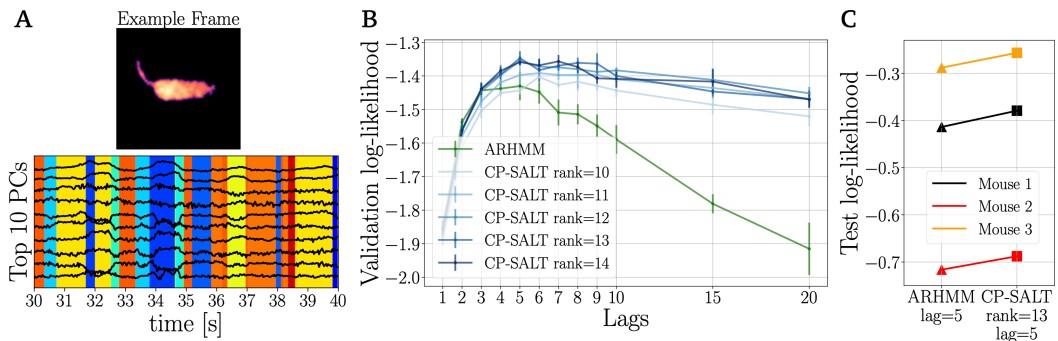

*Figure 4:* **CP-SALT consistently outperforms ARHMM on mouse behavior videos and segments data into distinct behavioral syllables**: **(A)** An example frame from the MoSeq dataset. The models were trained on the top 10 principal components of the video frames from three mice. **(B)** CP-SALT and ARHMM trained with different ranks and lags. Mean and standard deviation across five seeds evaluated on a validation set are shown. CP-SALT parameterization prevents overfitting for larger lags. **(C)** Test log-likelihood, averaged across 5 model fits, computed from the best ARHMM and CP-SALT hyperparameters in (B). CP-SALT outperforms ARHMM across all three mice.

### 5.4 Modeling *C. elegans* Neural Data

Finally, we analyzed neural recordings of an immobilized *C. elegans* worm from Kato et al. [2015]. SLDS have previously been used to capture the time-varying low-dimensional dynamics of the neural activity [Linderman et al., 2019, Glaser et al., 2020]. We compared SLDS, ARHMM, and CP-SALT with 18 minutes of neural traces (recorded at 3Hz; ∼3200 timesteps) from one worm, in which 48 neurons were confidently identified. The dataset also contains 7 manually identified state labels based on the neural activity.

We used $H = 7$ discrete states and fitted SLDSs, ARHMMs, and CP-SALT with varying lags and ranks (or continuous latent dimensions for SLDSs). Following Linderman et al. [2019], we searched for sets of hyperparameters that achieve ∼90% explained variance on a held-out test dataset (see Appendix F for more details). For ARHMMs and CP-SALT, we chose a larger lag ($L = 9$, equivalent to 3 seconds) to examine the long-timescale correlations among the neurons.

We find that SALT can perform as well as SLDSs and ARHMMs in terms of held-out explained variance ratio (a metric used by previous work [Linderman et al., 2019]). As expected, we find that CP-SALT can achieve these results with far fewer parameters than ARHMMs, and with a parameter count closer to SLDS than ARHMM (as more continuous latent states were required in an SLDS to achieve ∼90% explained variance; see Appendix F). Figure 5A shows that SALT, SLDS and ARHMM produce similar segmentations to the given labels, as evidenced by the confusion matrix having high entries on the leading diagonal (Figure 5B and Appendix F).

Figure 5C shows the one-dimensional autoregressive filters learned by CP-SALT, defined as $\sum_{i=1}^{D_1} \sum_{j=1}^{D_2} \sum_{k=1}^{D_3} g_{ijk}^{(h)} u_{pi}^{(h)} v_{qj}^{(h)} \mathbf{w}_{:k}^{(h)}$ for neurons $p$ and $q$. We see that neurons believed to be involved in particular behavioral states have high weights in the filter (e.g., SMDV during the "Ventral Turn" state and SMDD during the "Dorsal Turn" state [Linderman et al., 2019, Kato et al., 2015, Gray et al., 2005, Kaplan et al., 2020, Yeon et al., 2018]). This highlights how switching autoregressive models can reveal state-dependent functional interactions between neurons (or observed states more generally). In Appendix F, we show the autoregressive filters learned by an ARHMM, an SLDS, and a generalized linear model (GLM), a method commonly used to model inter-neuronal interactions [Pillow et al., 2008]. Interestingly, the GLM does not find many strong functional interactions between neurons, likely because it is averaging over many unique discrete states. In addition to its advantages in parameter efficiency and estimation, SALT thus provides a novel method for finding changing functional interactions across neurons at multiple timescales.

## 6 Discussion

We introduce switching autoregressive low-rank tensor (SALT) models: a novel model class that parameterizes the autoregressive tensors of an ARHMM with a low-rank factorization. This constraint

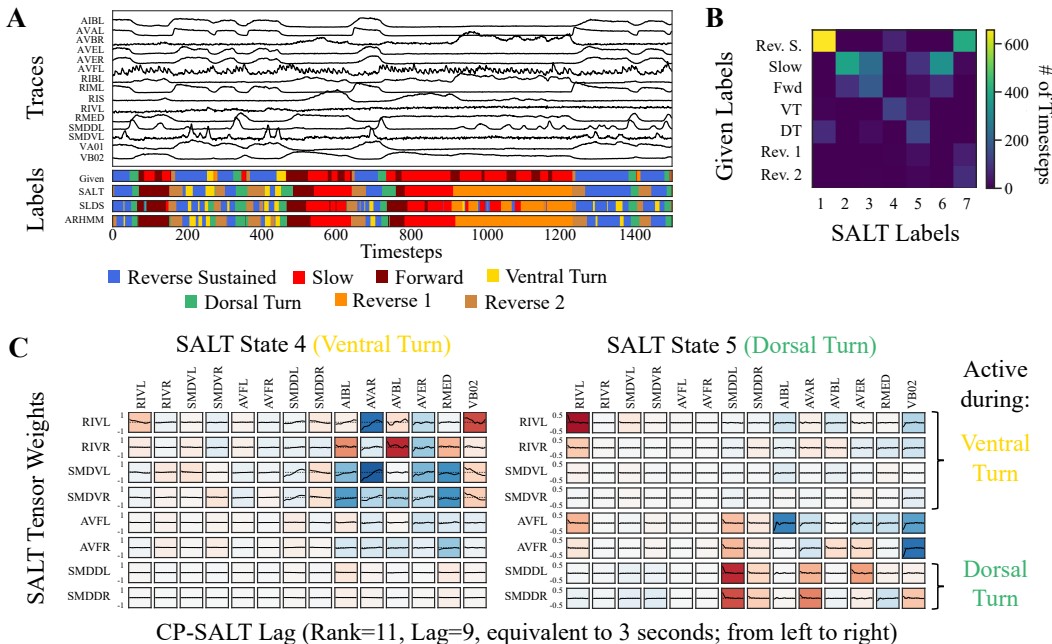

*Figure 5:* **CP-SALT provides good segmentations of *C. elegans* neural data, and inferred low-rank tensors give insights into temporal dependencies among neurons in each discrete state**: **(A)** Example data with manually generated labels (Given), as well as segmentations generated by SALT, SLDS and ARHMM models. Learned states are colored based on the permutation of states that best matches given labels. All methods produce comparable segmentations, with high agreement with the given labels. **(B)** Confusion matrix of SALT-generated labels. **(C)** One-dimensional autoregressive filters learned in two states by SALT (identified as ventral and dorsal turns). Colors indicate the area under curve (red is positive; blue is negative). The first four rows are neurons known to mediate ventral turns, while the last two rows mediate dorsal turns [Kato et al., 2015, Gray et al., 2005, Yeon et al., 2018]. These known behavior-tuned neurons generally have larger magnitude autoregressive filters. Interestingly, AVFL and AVFR also have large filters for dorsal turns. These neurons do not have a well-known function. However, they are associated with motor neurons, and so may simultaneously activate due to factors that co-occur with turning. This highlights how SALT may be used for proposing novel relationships in systems.

allows SALT to model time-series data with fewer parameters than ARHMMs and with simpler estimation procedures than SLDSs. We also make theoretical connections between low-rank tensor regressions and LDSs. We then demonstrate, with both synthetic and real datasets, that SALT offers both efficiency and interpretability, striking an advantageous balance between the ARHMM and SLDS. Moreover, SALT offers an enhanced ability to investigate the interactions across observations, such as neurons, across different timescales in a data-efficient manner.

However, SALT is not without limitations. Foremost, SALT cannot readily handle missing observations, or share information between multiple time series with variable observation dimensions. "Hierarchical SALT" is an interesting extension, where information is shared across time series, but the factors of individual time series are allowed to vary. Furthermore, SALT could be extended to handle non-Gaussian data. For example, neural spike trains are often modeled with Poisson likelihoods instead of SALT's Gaussian noise model. In this case, the E-step would still be exact, but the M-step would no longer have closed-form coordinate updates. Despite these limitations, SALT offers simple, effective and complementary means of modeling and inference methodology for complex, time-varying dynamical systems.

**Ethical Concerns** We note that there are no new ethical concerns as a result of SALT.

## Acknowledgments and Disclosure of Funding

This work was supported by grants from the Simons Collaboration on the Global Brain (SCGB 697092), the NIH (U19NS113201, R01NS113119, R01NS130789, and K99NS119787), the Sloan Foundation, and the Stanford Center for Human and Artificial Intelligence. We thank Liam Paninski and the anonymous reviewers for their constructive feedback on the paper. We also thank the members of the Linderman Lab for their support and feedback throughout the project.

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

# Supplementary Materials for: Switching Autoregressive Low-rank Tensor Models

**Table of Contents**

# A  SALT Optimization via Tensor Regression

Let $\mathbf{y}_t \in \mathbb{R}^{N_1}$ be the $t$-th outputs and $\mathbf{X}_t \in \mathbb{R}^{N_2 \times N_3}$ be the $t$-th inputs. The regression weights are a tensor $\mathcal{A} \in \mathbb{R}^{N_1 \times N_2 \times N_3}$, which we model via a Tucker decomposition,

$$\mathcal{A} = \sum_{i=1}^{D_1} \sum_{j=1}^{D_2} \sum_{k=1}^{D_3} g_{ijk}\, \mathbf{u}_{:i} \circ \mathbf{v}_{:j} \circ \mathbf{w}_{:k}, \tag{11}$$

where $\mathbf{u}_i$, $\mathbf{v}_j$, and $\mathbf{w}_k$ are columns of the factor matrices $\mathbf{U} \in \mathbb{R}^{N_1 \times D_1}$, $\mathbf{V} \in \mathbb{R}^{N_2 \times D_2}$, and $\mathbf{W} \in \mathbb{R}^{N_3 \times D_3}$, respectively, and $g_{ijk}$ are entries in the core tensor $\mathcal{G} \in \mathbb{R}^{D_1 \times D_2 \times D_3}$.

We define $\times_{j,k}$ to be a tensor-matrix product over the $j^{th}$ and $k^{th}$ slices of the tensor. For example, given a three-way tensor $\mathcal{A} \in \mathbb{R}^{D_1 \times D_2 \times D_3}$ and a matrix $\mathbf{X} \in \mathbb{R}^{D_2 \times D_3}$, $\mathcal{A} \times_{2,3} \mathbf{X} = \sum_{j=1}^{D_2} \sum_{k=1}^{D_3} \mathbf{a}_{:jk} x_{jk}$. This operation is depicted in Figure 6.

Consider the linear model, $\mathbf{y}_t \sim \mathcal{N}(\mathcal{A} \times_{2,3} \mathbf{X}_t, \mathbf{Q})$ where $\mathcal{A} \times_{2,3} \mathbf{X}_t$ is defined using the Tucker decomposition of $\mathcal{A}$ as,

$$\mathcal{A} \times_{2,3} \mathbf{X}_t = \mathcal{A}_{(1)} \mathrm{vec}(\mathbf{X}_t) \tag{12}$$

$$= \mathbf{U}\mathcal{G}_{(1)}(\mathbf{V}^\top \otimes \mathbf{W}^\top)\mathrm{vec}(\mathbf{X}_t) \tag{13}$$

$$= \mathbf{U}\mathcal{G}_{(1)}\mathrm{vec}(\mathbf{V}^\top \mathbf{X}_t \mathbf{W}) \tag{14}$$

where $\mathcal{A}_{(1)} \in \mathbb{R}^{N_1 \times N_2 N_3}$ and $\mathcal{G}_{(1)} \in \mathbb{R}^{D_1 \times D_2 D_3}$ are mode-1 matricizations of the corresponding tensors. *Note that these equations assume that matricization and vectorization are performed in row-major order, as in Python but opposite to what is typically used in Wikipedia articles.*

Equation (14) can be written in multiple ways, and these equivalent forms will be useful for deriving the updates below. We have,

$$\mathcal{A} \times_{2,3} \mathbf{X}_t = \mathbf{U}\mathcal{G}_{(1)}(\mathbf{I}_{D_2} \otimes \mathbf{W}^\top \mathbf{X}_t^\top)\mathrm{vec}(\mathbf{V}^\top) \tag{15}$$

$$= \mathbf{U}\mathcal{G}_{(1)}(\mathbf{V}^\top \mathbf{X}_t \otimes \mathbf{I}_{D_3})\mathrm{vec}(\mathbf{W}) \tag{16}$$

$$= \left[\mathbf{U} \otimes \mathrm{vec}(\mathbf{V}^\top \mathbf{X}_t \mathbf{W})\right] \mathrm{vec}(\mathcal{G}). \tag{17}$$

We minimize the negative log likelihood by coordinate descent.

**Optimizing the output factors**  Let

$$\widetilde{\mathbf{x}}_t = \mathcal{G}_{(1)}\mathrm{vec}(\mathbf{V}^\top \mathbf{X}_t \mathbf{W}) \tag{18}$$

for fixed $\mathbf{V}$, $\mathbf{W}$, and $\mathcal{G}$. The NLL as a function of $\mathbf{U}$ is,

$$\mathcal{L}(\mathbf{U}) = \frac{1}{2}\sum_t (\mathbf{y}_t - \mathbf{U}\widetilde{\mathbf{x}}_t)^\top \mathbf{Q}^{-1}(\mathbf{y}_t - \mathbf{U}\widetilde{\mathbf{x}}_t). \tag{19}$$

This is a standard least squares problem with solution

$$\mathbf{U}^\star = \left(\sum_t \mathbf{y}_t \widetilde{\mathbf{x}}_t^\top\right)\left(\sum_t \widetilde{\mathbf{x}}_t \widetilde{\mathbf{x}}_t^\top\right)^{-1}. \tag{20}$$

**Optimizing the core tensors**  Let $\widetilde{\mathbf{X}}_t = \mathbf{U} \otimes \mathrm{vec}(\mathbf{V}^\top \mathbf{X}_t \mathbf{W}) \in \mathbb{R}^{N_1 \times D_1 D_2 D_3}$ denote the coefficient on $\mathrm{vec}(\mathcal{G})$ in eq. (17). The NLL as a function of $\mathbf{g} = \mathrm{vec}(\mathcal{G})$ is,

$$\mathcal{L}(\mathbf{g}) = \frac{1}{2}\sum_t (\mathbf{y}_t - \widetilde{\mathbf{X}}_t \mathbf{g})^\top \mathbf{Q}^{-1}(\mathbf{y}_t - \widetilde{\mathbf{X}}_t \mathbf{g}). \tag{21}$$

The minimizer of this quadratic form is,

$$\mathbf{g}^\star = \left(\sum_t \widetilde{\mathbf{X}}_t^\top \mathbf{Q}^{-1}\widetilde{\mathbf{X}}_t\right)^{-1}\left(\sum_t \widetilde{\mathbf{X}}_t^\top \mathbf{Q}^{-1}\mathbf{y}_t\right) \tag{22}$$

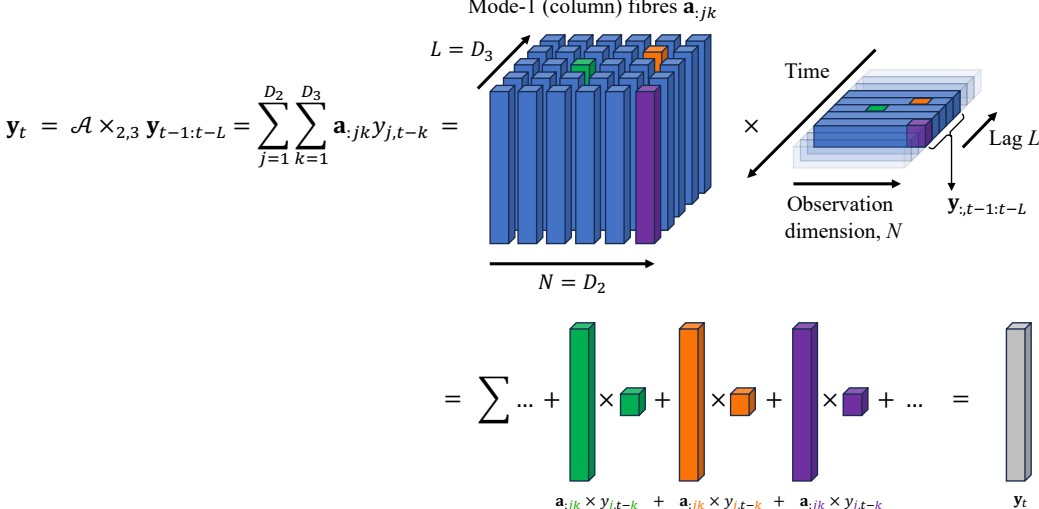

$$\mathbf{y}_t \;=\; \mathcal{A} \times_{2,3} \mathbf{y}_{t-1:t-L} = \sum_{j=1}^{D_2}\sum_{k=1}^{D_3} \mathbf{a}_{:jk} y_{j,t-k} \;=$$

*Figure 6:* Depiction of the $\times_{2,3}$ tensor operator we use [Kolda and Bader, 2009] This can be thought of as a generalization of matrix-vector products to tensor-matrix products.

**Optimizing the input factors**   Let

$$\widetilde{\mathbf{X}}_t = \mathbf{U}\mathcal{G}_{(1)}(\mathbf{I}_{D_2}\otimes\mathbf{W}^\top\mathbf{X}_t^\top) \tag{23}$$

for fixed $\mathbf{U}$, $\mathbf{W}$, and $\mathcal{G}$. The NLL as a function of $\mathbf{v}=\mathrm{vec}(\mathbf{V}^\top)$ is,

$$\mathcal{L}(\mathbf{v}) = \frac{1}{2}\sum_t (\mathbf{y}_t - \widetilde{\mathbf{X}}_t\mathbf{v})^\top \mathbf{Q}^{-1}(\mathbf{y}_t - \widetilde{\mathbf{X}}_t\mathbf{v}). \tag{24}$$

The minimizer of this quadratic form is,

$$\mathbf{v}^\star = \left(\sum_t \widetilde{\mathbf{X}}_t^\top \mathbf{Q}^{-1}\widetilde{\mathbf{X}}_t\right)^{-1}\left(\sum_t \widetilde{\mathbf{X}}_t^\top \mathbf{Q}^{-1}\mathbf{y}_t\right) \tag{25}$$

**Optimizing the lag factors**   Let

$$\widetilde{\mathbf{X}}_t = \mathbf{U}\mathcal{G}_{(1)}(\mathbf{V}^\top\mathbf{X}_t\otimes\mathbf{I}_{D_3}) \tag{26}$$

for fixed $\mathbf{U}$, $\mathbf{V}$, and $\mathcal{G}$. The NLL as a function of $\mathbf{w}=\mathrm{vec}(\mathbf{W})$ is,

$$\mathcal{L}(\mathbf{w}) = \frac{1}{2}\sum_t (\mathbf{y}_t - \widetilde{\mathbf{X}}_t\mathbf{w})^\top \mathbf{Q}^{-1}(\mathbf{y}_t - \widetilde{\mathbf{X}}_t\mathbf{w}). \tag{27}$$

The minimizer of this quadratic form is,

$$\mathbf{w}^\star = \left(\sum_t \widetilde{\mathbf{X}}_t^\top \mathbf{Q}^{-1}\widetilde{\mathbf{X}}_t\right)^{-1}\left(\sum_t \widetilde{\mathbf{X}}_t^\top \mathbf{Q}^{-1}\mathbf{y}_t\right) \tag{28}$$

**Multiple discrete states**   If we have discrete states $z_t \in \{1,\dots,H\}$ and each state has its own parameters $(\mathcal{G}^{(h)}, \mathbf{U}^{(h)}, \mathbf{V}^{(h)}, \mathbf{W}^{(h)}, \mathbf{Q}^{(h)})$, then letting $\omega_t^{(h)} = \mathbb{E}[z_t = h]$ denote the weights from the E-step, the summations in coordinate updates are weighted by $\omega_t^{(h)}$. For example, the coordinate update for the core tensors becomes,

$$\mathbf{g}^{(h)\star} = \left(\sum_t \omega_t^{(h)}\widetilde{\mathbf{X}}_t^{(h)\top} \mathbf{Q}^{(h)-1}\widetilde{\mathbf{X}}_t^{(h)}\right)^{-1}\left(\sum_t \omega_t^{(h)}\widetilde{\mathbf{X}}_t^{(h)\top} \mathbf{Q}^{(h)-1}\mathbf{y}_t\right) \tag{29}$$

# B SALT approximates a (Switching) Linear Dynamical System

We now re-state and provide a full proof for Proposition 1.

**Proposition 1** (Low-Rank Tensor Autoregressions Approximate Stable Linear Dynamical Systems).
*Consider a stable linear time-invariant Gaussian dynamical system. We define the steady-state Kalman gain matrix as $\mathbf{K} = \lim_{t \to \infty} \mathbf{K}_t$, and $\boldsymbol{\Gamma} = \mathbf{A}(\mathbf{I} - \mathbf{KC})$. The matrix $\boldsymbol{\Gamma} \in \mathbb{R}^{S \times S}$ has eigenvalues $\lambda_1, \ldots, \lambda_S$. Let $\lambda_{\max} = \max_s |\lambda_s|$; for a stable LDS, $\lambda_{\max} < 1$ [Davis and Vinter, 1985]. Let $n$ denote the number of real eigenvalues and $m$ the number of complex conjugate pairs. Let $\hat{\mathbf{y}}_t^{(\mathsf{LDS})} = \mathbb{E}[\mathbf{y}_t \mid \mathbf{y}_{1:t-1}]$ denote the predictive mean under a steady-state LDS, and $\hat{\mathbf{y}}_t^{(\mathsf{SALT})}$ the predictive mean under a SALT model. An order-$L$ Tucker-SALT model with rank $n + 2m = S$, or a CP-SALT model with rank $n + 3m$, can approximate the predictive mean of the steady-state LDS with error $\|\hat{\mathbf{y}}_t^{(\mathsf{LDS})} - \hat{\mathbf{y}}_t^{(\mathsf{SALT})}\|_\infty = \mathcal{O}(\lambda_{\max}^L)$.*

*Proof.* A stationary linear dynamical system (LDS) is defined as follows:

$$\mathbf{x}_t = \mathbf{A}\mathbf{x}_{t-1} + \mathbf{b} + \boldsymbol{\epsilon}_t \tag{30}$$

$$\mathbf{y}_t = \mathbf{C}\mathbf{x}_t + \mathbf{d} + \boldsymbol{\delta}_t \tag{31}$$

where $\mathbf{y}_t \in \mathbb{R}^N$ is the $t$-th observation, $\mathbf{x}_t \in \mathbb{R}^S$ is the $t$-th hidden state, $\boldsymbol{\epsilon}_t \overset{\text{i.i.d.}}{\sim} \mathcal{N}(\mathbf{0}, \mathbf{Q})$, $\boldsymbol{\delta}_t \overset{\text{i.i.d.}}{\sim} \mathcal{N}(\mathbf{0}, \mathbf{R})$, and $\boldsymbol{\theta} = (\mathbf{A}, \mathbf{b}, \mathbf{Q}, \mathbf{C}, \mathbf{d}, \mathbf{R})$ are the parameters of the LDS.

Following the notation of Murphy [2012], the one-step-ahead posterior predictive distribution for the observations of the LDS defined above can be expressed as:

$$p(\mathbf{y}_t|\mathbf{y}_{1:t-1}) = \mathcal{N}(\mathbf{C}\boldsymbol{\mu}_{t|t-1} + \mathbf{d}, \mathbf{C}\boldsymbol{\Sigma}_{t|t-1}\mathbf{C}^T + \mathbf{R}) \tag{32}$$

where

$$\boldsymbol{\mu}_{t|t-1} = \mathbf{A}\boldsymbol{\mu}_{t-1} + \mathbf{b} \tag{33}$$

$$\boldsymbol{\mu}_t = \boldsymbol{\mu}_{t|t-1} + \mathbf{K}_t\mathbf{r}_t \tag{34}$$

$$\boldsymbol{\Sigma}_{t|t-1} = \mathbf{A}\boldsymbol{\Sigma}_{t-1}\mathbf{A}^T + \mathbf{Q} \tag{35}$$

$$\boldsymbol{\Sigma}_t = (\mathbf{I} - \mathbf{K}_t\mathbf{C})\boldsymbol{\Sigma}_{t|t-1} \tag{36}$$

$$p(\mathbf{x}_1) = \mathcal{N}(\mathbf{x}_1 \mid \boldsymbol{\mu}_{1|0}, \boldsymbol{\Sigma}_{1|0}) \tag{37}$$

$$\mathbf{K}_t = (\boldsymbol{\Sigma}_{t|t-1}^{-1} + \mathbf{C}^T\mathbf{R}\mathbf{C})^{-1}\mathbf{C}^T\mathbf{R}^{-1} \tag{38}$$

$$\mathbf{r}_t = \mathbf{y}_t - \mathbf{C}\boldsymbol{\mu}_{t|t-1} - \mathbf{d}. \tag{39}$$

We can then expand the mean $\mathbf{C}\boldsymbol{\mu}_{t|t-1} + \mathbf{d}$ as follows:

$$\mathbf{C}\boldsymbol{\mu}_{t|t-1} + \mathbf{d} = \mathbf{C}\sum_{l=1}^{t-1} \boldsymbol{\Gamma}_l \mathbf{A}\mathbf{K}_{t-l}\mathbf{y}_{t-l} + \mathbf{C}\sum_{l=1}^{t-1} \boldsymbol{\Gamma}_l(\mathbf{b} - \mathbf{A}\mathbf{K}_{t-l}\mathbf{d}) + \mathbf{d} \tag{40}$$

where

$$\boldsymbol{\Gamma}_l = \prod_{i=1}^{l-1} \mathbf{A}(\mathbf{I} - \mathbf{K}_{t-i}\mathbf{C}) \quad \text{for} \quad l \in \{2, 3, \ldots\}, \tag{41}$$

$$\boldsymbol{\Gamma}_1 = \mathbf{I}. \tag{42}$$

Theorem 3.3.3 of Davis and Vinter [1985] (reproduced with our notation below) states that for a stabilizable and detectable system, the $\lim_{t\to\infty}\boldsymbol{\Sigma}_{t|t-1} = \boldsymbol{\Sigma}$, where $\boldsymbol{\Sigma}$ is the unique solution of the discrete algebraic Riccati equation

$$\boldsymbol{\Sigma} = \mathbf{A}\boldsymbol{\Sigma}\mathbf{A}^T - \mathbf{A}\boldsymbol{\Sigma}\mathbf{C}^T(\mathbf{C}\boldsymbol{\Sigma}\mathbf{C}^T + \mathbf{R})^{-1}\mathbf{C}\boldsymbol{\Sigma}\mathbf{A}^T + \mathbf{Q}. \tag{43}$$

As we are considering stable autonomous LDSs here, the system is stabilizable and detectable, as all unobservable states are themselves stable [Davis and Vinter, 1985, Katayama, 2005]

**Theorem 3.3.3** (Reproduced from Davis and Vinter [1985], updated to our notation and context).
*The theorem has two parts.*

(a) *If the pair $(\mathbf{A}, \mathbf{C})$ is detectable then there exists at least one non-negative solution, $\boldsymbol{\Sigma}$, to the discrete algebraic Riccati equation (43).*

(b) *If the pair $(\mathbf{A}, \mathbf{C})$ is stabilizable then this solution $\boldsymbol{\Sigma}$ is unique, and $\boldsymbol{\Sigma}_{t|t-1} \to \boldsymbol{\Sigma}$ as $t \to \infty$, where $\boldsymbol{\Sigma}_{t|t-1}$ is the sequence generated by (33)-(39) with arbitrary initial covariance $\boldsymbol{\Sigma}_0$. Then, the matrix $\boldsymbol{\Gamma} = \mathbf{A}(\mathbf{I} - \mathbf{KC})$ is stable, where $\mathbf{K}$ is the Kalman gain corresponding to $\boldsymbol{\Sigma}$; i.e.,*

$$\mathbf{K} = (\boldsymbol{\Sigma}^{-1} + \mathbf{C}^T \mathbf{R} \mathbf{C})^{-1} \mathbf{C}^T \mathbf{R}^{-1} \tag{44}$$

*Proof.* See Davis and Vinter [1985]. Note that Davis and Vinter [1985] define the Kalman gain as $\mathbf{A}\mathbf{K}$. □

The convergence of the Kalman gain also implies that each term in the sequence $\boldsymbol{\Gamma}_l$ converges to

$$\boldsymbol{\Gamma}_l = \prod_{i=1}^{l-1} \mathbf{A}(\mathbf{I} - \mathbf{KC}) = (\mathbf{A}(\mathbf{I} - \mathbf{KC}))^{l-1} = \boldsymbol{\Gamma}^{l-1}, \tag{45}$$

where, concretely, we define $\boldsymbol{\Gamma} = \mathbf{A}(\mathbf{I} - \mathbf{KC})$. We can therefore make the following substitution and approximation

$$\mathbf{C}\boldsymbol{\mu}_{t|t-1} + \mathbf{d} \overset{\lim t \to \infty}{=} \mathbf{C}\sum_{l=1}^{t-1} \boldsymbol{\Gamma}^l \mathbf{A}\mathbf{K}\mathbf{y}_{t-l} + \mathbf{C}\sum_{l=1}^{t-1} \boldsymbol{\Gamma}^l (\mathbf{b} - \mathbf{A}\mathbf{K}\mathbf{d}) + \mathbf{d} \tag{46}$$

$$= \mathbf{C}\sum_{l=1}^{L} \boldsymbol{\Gamma}^l \mathbf{A}\mathbf{K}\mathbf{y}_{t-l} + \mathbf{C}\sum_{l=1}^{L} \boldsymbol{\Gamma}^l (\mathbf{b} - \mathbf{A}\mathbf{K}\mathbf{d}) + \mathbf{d} + \sum_{l=L+1}^{\infty} \mathcal{F}\left(\boldsymbol{\Gamma}^l\right) \tag{47}$$

$$\approx \mathbf{C}\sum_{l=1}^{L} \boldsymbol{\Gamma}^l \mathbf{A}\mathbf{K}\mathbf{y}_{t-l} + \mathbf{C}\sum_{l=1}^{L} \boldsymbol{\Gamma}^l (\mathbf{b} - \mathbf{A}\mathbf{K}\mathbf{d}) + \mathbf{d} \tag{48}$$

The approximation is introduced as a result of truncating the sequence to consider just the "first" $L$ terms, and discarding the higher-order terms (indicated in blue). It is important to note that each term in (46) is the sum of a geometric sequence multiplied elementwise with $\mathbf{y}_t$.

There are two components we prove from here. First, we derive an element-wise bound on the error introduced by the truncation, and verify that under the conditions outlined that the bound decays monotonically in $L$. We then show that Tucker and CP decompositions can represent the truncated summations in (48), and derive the minimum rank required for this representation to be exact.

**Bounding The Error Term** We first rearrange the truncated terms in (46), where we define $\mathbf{x}_l \triangleq \mathbf{A}\mathbf{K}\mathbf{y}_{t-l} + \mathbf{b} - \mathbf{A}\mathbf{K}\mathbf{d}$

$$\sum_{l=L+1}^{\infty} \mathcal{F}\left(\boldsymbol{\Gamma}^l\right) = \mathbf{C}\sum_{l=L+1}^{\infty} \boldsymbol{\Gamma}^l \mathbf{A}\mathbf{K}\mathbf{y}_{t-l} + \mathbf{C}\sum_{l=L+1}^{\infty} \boldsymbol{\Gamma}^l (\mathbf{b} - \mathbf{A}\mathbf{K}\mathbf{d}) + \mathbf{d}, \tag{49}$$

$$= \sum_{l=L+1}^{\infty} \mathbf{C}\boldsymbol{\Gamma}^l \mathbf{x}_l, \tag{50}$$

$$= \sum_{l=L+1}^{\infty} \mathbf{C}\mathbf{E}\boldsymbol{\Lambda}^{l-1}\mathbf{E}^{-1}\mathbf{x}_l, \tag{51}$$

$$= \sum_{l=L+1}^{\infty} \mathbf{P}\boldsymbol{\Lambda}^{l-1}\mathbf{q}_l, \tag{52}$$

where $\mathbf{E}\boldsymbol{\Lambda}\mathbf{E}^{-1}$ is the eigendecomposition of $\boldsymbol{\Gamma}$, $\mathbf{P} \triangleq \mathbf{C}\mathbf{E}$, and $\mathbf{q}_l \triangleq \mathbf{E}^{-1}\mathbf{x}_l$. We now consider the infinity-norm of the error, and apply the triangle and Cauchy-Schwartz inequalities. We can write the

bound on the as

$$\epsilon = \left| \left( \sum_{l=L+1}^{\infty} \mathcal{F}\left(\mathbf{\Gamma}^l\right) \right)_n \right|, \quad \text{where} \quad n = \arg\max_k \left| \left( \sum_{l=L+1}^{\infty} \mathcal{F}\left(\mathbf{\Gamma}^l\right) \right)_k \right| \tag{53}$$

$$= \left| \sum_{l=L+1}^{\infty} \sum_{s=1}^{S} p_{ns} \lambda_d^{l-1} q_{l,s} \right|, \tag{54}$$

$$\leq \sum_{l=L+1}^{\infty} \sum_{s=1}^{S} |p_{ns}| \left| \lambda_s^{l-1} \right| |q_{l,s}|. \tag{55}$$

Upper bounding the absolute magnitude of $q_{l,s}$ by $W$ provides a further upper bound, which we can then rearrange

$$\epsilon \leq W \sum_{l=L+1}^{\infty} \sum_{s=1}^{S} |p_{ns}| \left| \lambda_s^{l-1} \right|, \tag{56}$$

$$= W \sum_{s=1}^{S} |p_{ns}| \sum_{l=L+1}^{\infty} \left| \lambda_s^{l-1} \right|. \tag{57}$$

The first two terms are constant, and hence the upper bound is determined by the sum of the of the $l^{\text{th}}$ power of the eigenvalues. We can again bound this sum by setting all eigenvalues equal to the magnitude of the eigenvalue with the maximum magnitude (spectral norm), denoted $\lambda_{max}$:

$$\epsilon \leq W \sum_{s=1}^{S} |p_{ns}| \sum_{l=L+1}^{\infty} \lambda_{\max}^{l-1}, \tag{58}$$

where these second summation is not a function of $s$, and $W \sum_{s=1}^{S} |p_{ns}|$ is constant. This summation is a truncated geometric sequence. Invoking Theorem 3.3.3 of Davis and Vinter [1985] again, the matrix $\mathbf{\Gamma}$ has only stable eigenvalues, and hence $\lambda_{\max} < 1$. Therefore the sequence sum will converge to

$$\sum_{l=L+1}^{\infty} \lambda_{\max}^{l-1} = \frac{\lambda_{\max}^L}{1 - \lambda_{\max}}. \tag{59}$$

Rearranging again, we see that the absolute error on the $n^{\text{th}}$ element of $\mathbf{y}_t$ is therefore bounded according to a power of the spectral norm

$$\epsilon \leq W \sum_{s=1}^{S} |p_{ns}| \frac{\lambda_{\max}^L}{1 - \lambda_{\max}}, \tag{60}$$

$$= \mathcal{O}\left(\lambda_{\max}^L\right). \tag{61}$$

More specifically, for a stable linear time-invariant dynamical system, and where $\mathbf{q}$ — and hence $\mathbf{y}$ — is bounded, then the bound on the error incurred reduces exponentially in the length of the window $L$. Furthermore, this error bound will reduce faster for systems with a lower spectral norm.

**Diagonalizing the System** We first transform $\mathbf{\Gamma}$ into real modal form, defined as $\mathbf{E}\mathbf{\Lambda}\mathbf{E}^{-1}$, where $\mathbf{E}$ and $\mathbf{\Lambda}$ are the eigenvectors and diagonal matrix of eigenvalues of $\mathbf{\Gamma}$. Letting $\mathbf{\Gamma}$ have $n$ real eigenvalues and $m$ pairs of complex eigenvalues (i.e., $n + 2m = S$), we can express $\mathbf{E}$, $\mathbf{\Lambda}$, and $\mathbf{E}^{-1}$ as:

$$\mathbf{E} = \begin{bmatrix} \mathbf{a}_1 & \dots & \mathbf{a}_n & \mathbf{b}_1 & \mathbf{c}_1 & \dots & \mathbf{b}_m & \mathbf{c}_m \end{bmatrix} \tag{62}$$

$$\mathbf{\Lambda} = \begin{bmatrix} \lambda_1 & & & & & & & \\ & \ddots & & & & & & \\ & & \lambda_n & & & & & \\ & & & \sigma_1 & \omega_1 & & & \\ & & & -\omega_1 & \sigma_1 & & & \\ & & & & & \ddots & & \\ & & & & & & \sigma_m & \omega_m \\ & & & & & & -\omega_m & \sigma_m \end{bmatrix} \tag{63}$$

$$\mathbf{E}^{-1} = \begin{bmatrix} \mathbf{d}_1^T \\ \vdots \\ \mathbf{d}_n^T \\ \mathbf{e}_1^T \\ \boldsymbol{f}_1^T \\ \vdots \\ \mathbf{e}_m^T \\ \boldsymbol{f}_m^T \end{bmatrix} \tag{64}$$

where $\mathbf{a}_1 \dots \mathbf{a}_n$ are the right eigenvectors corresponding to $n$ real eigenvalues $\lambda_1 \dots \lambda_n$, and $\mathbf{b}_i$ and $\mathbf{c}_i$ are the real and imaginary parts of the eigenvector corresponding to the complex eigenvalue $\sigma_i + j\omega_i$. Note that

$$\boldsymbol{\Gamma}^l = (\mathbf{A}(\mathbf{I} - \mathbf{KC}))^{l-1} = \mathbf{E}\boldsymbol{\Lambda}^{l-1}\mathbf{E}^{-1} \tag{65}$$

The $l^{th}$ power of $\boldsymbol{\Lambda}$, $\boldsymbol{\Lambda}^l$, where $l \geq 0$, can be expressed as:

$$\boldsymbol{\Lambda}^l = \begin{bmatrix} \lambda_1^l & & & & & & & \\ & \ddots & & & & & & \\ & & \lambda_n^l & & & & & \\ & & & \sigma_{1,l} & \omega_{1,l} & & & \\ & & & -\omega_{1,l} & \sigma_{1,l} & & & \\ & & & & & \ddots & & \\ & & & & & & \sigma_{m,l} & \omega_{m,l} \\ & & & & & & -\omega_{m,l} & \sigma_{m,l} \end{bmatrix} \tag{66}$$

where $\sigma_{i,l} = \sigma_{i,l-1}^2 - \omega_{i,l-1}^2$, $\omega_{i,l} = 2\sigma_{i,l-1}\omega_{i,l-1}$ for $l \geq 2$, $\sigma_{i,1} = \sigma_i$, $\omega_{i,1} = \omega_i$, $\sigma_{i,0} = 1$, and $\omega_{i,0} = 0$.

**Tucker Tensor Regression**    Let $\mathcal{H} \in \mathbb{R}^{S \times S \times L}$ be a three-way tensor, whose $l^{th}$ frontal slice $\mathbf{H}_{::l} = \boldsymbol{\Lambda}^{l-1}$. Let $\mathcal{G} \in \mathbb{R}^{S \times S \times S}$ be a three-way tensor, whose entry $g_{ijk} = \mathbb{1}_{i=j=k}$ for $1 \leq k \leq n$, and $g_{ijk} = (-1)^{\mathbb{1}_{i+1=j=k+1}} \mathbb{1}_{(i=j=k)\vee(i-1=j-1=k)\vee(i=j+1=k+1)\vee(i+1=j=k+1)}$ for $k \in \{n+1, n+3, \dots, n+2m-1\}$. Let $\mathbf{W} \in \mathbb{R}^{L \times S}$ be a matrix, whose entry $w_{lk} = \lambda_k^{l-1}$ for $1 \leq k \leq n$, $w_{lk} = \sigma_{k,l-1}$ for $k \in \{n+1, n+3, \dots, n+2m-1\}$, and $w_{lk} = -\omega_{k,l-1}$ for $k \in \{n+2, n+4, \dots, n+2m\}$. We can then decompose $\mathcal{H}$ into $\mathcal{G} \in \mathbb{R}^{S \times S \times S}$ and $\mathbf{W} \in \mathbb{R}^{L \times S}$ such that $\mathcal{H} = \mathcal{G} \times_3 \mathbf{W}$ (Figure 7).

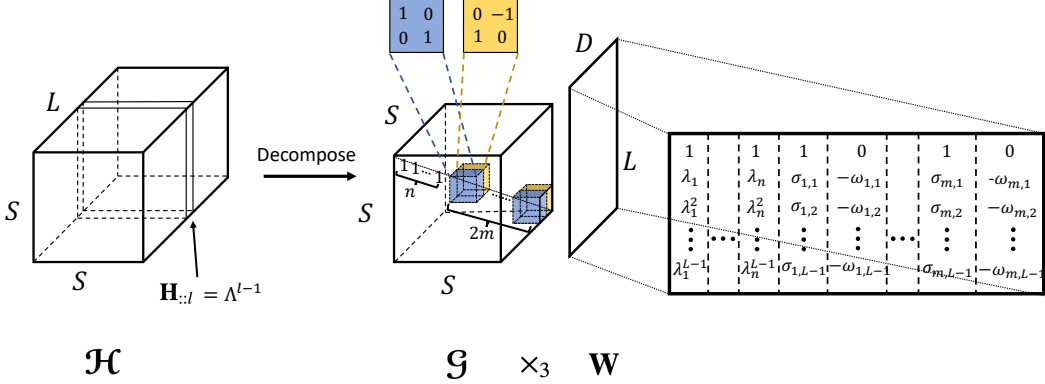

*Figure 7:* **Decomposition of $\mathcal{H}$ into $\mathcal{G}$ and $\mathbf{W}$ such that $\mathcal{H} = \mathcal{G} \times_3 \mathbf{W}$:** Given an LDS whose $\mathbf{A}(\mathbf{I} - \mathbf{KC})$ has $n$ real eigenvalues and $m$ pairs of complex eigenvalues, this decomposition illustrates how Tucker-SALT can approximate the LDS well with rank $n + 2m$.

With $\mathbf{V} = (\mathbf{E}^{-1}\mathbf{AK})^T$, $\mathbf{U} = \mathbf{CE}$, $\mathbf{m} = \mathbf{C}\sum_{l=1}^{L} \mathbf{\Gamma}^l(\mathbf{b} - \mathbf{AKd}) + \mathbf{d}$, and $\mathbf{X}_t = \mathbf{y}_{t-1:t-L}$, we can rearrange the mean to:

$$\mathbf{C}\boldsymbol{\mu}_{t|t-1} + \mathbf{d} \approx \mathbf{C}\sum_{l=1}^{L} \mathbf{E}\mathbf{\Lambda}^{l-1}\mathbf{E}^{-1}\mathbf{AK}\mathbf{y}_{t-l} + \mathbf{C}\sum_{l=1}^{L} \mathbf{\Gamma}^l(\mathbf{b} - \mathbf{AKd}) + \mathbf{d} \tag{67}$$

$$= \mathbf{U}\sum_{l=1}^{L} \mathbf{H}_{::l}\mathbf{V}^T\mathbf{y}_{t-l} + \mathbf{m} \tag{68}$$

$$= \mathbf{U}\sum_{l=1}^{L} (\mathcal{G}\bar{\times}_3\mathbf{w}_l)\mathbf{V}^T\mathbf{y}_{t-l} + \mathbf{m} \tag{69}$$

$$= \mathbf{U}\sum_{l=1}^{L} ((\mathcal{G} \times_2 \mathbf{V})\bar{\times}_3\mathbf{w}_l)\mathbf{y}_{t-l} + \mathbf{m} \tag{70}$$

$$= \mathbf{U}\sum_{l=1}^{L}\sum_{j=1}^{S}\sum_{k=1}^{S} \mathbf{g}_{:jk} \circ \mathbf{v}_{:j}(w_{lk}\mathbf{y}_{t-l}) + \mathbf{m} \tag{71}$$

$$= \mathbf{U}\sum_{j=1}^{S}\sum_{k=1}^{S} \mathbf{g}_{:jk}(\mathbf{v}_{:j}^\top\mathbf{X}_t\mathbf{w}_{:k}) + \mathbf{m} \tag{72}$$

$$= \sum_{i=1}^{S}\sum_{j=1}^{S}\sum_{k=1}^{S} \mathbf{u}_{:i}g_{ijk}(\mathbf{v}_{:j}^\top\mathbf{X}_t\mathbf{w}_{:k}) + \mathbf{m} \tag{73}$$

$$= \left[\sum_{i=1}^{n+2m}\sum_{j=1}^{n+2m}\sum_{k=1}^{n+2m} g_{ijk}\mathbf{u}_{:i} \circ \mathbf{v}_{:j} \circ \mathbf{w}_{:k}\right] \times_{2,3} \mathbf{X}_t + \mathbf{m} \tag{74}$$

**CP Tensor Regression** By rearranging $\mathbf{E}$, $\mathbf{\Lambda}^l$, and $\mathbf{E}^{-1}$ into $\mathbf{J}$, $\mathbf{P}_l$, and $\mathbf{S}$ respectively as follows:

$$\mathbf{J} = [\, \mathbf{a}_1 \; \ldots \; \mathbf{a}_n \; \mathbf{b}_1 + \mathbf{c}_1 \; \mathbf{b}_1 \; \mathbf{c}_1 \; \ldots \; \mathbf{b}_m + \mathbf{c}_m \; \mathbf{b}_m \; \mathbf{c}_m \,] \tag{75}$$

$$
\mathbf{P}_l = \begin{bmatrix}
\lambda_1^l & & & & & & & & & \\
& \ddots & & & & & & & & \\
& & \lambda_n^l & & & & & & & \\
& & & \sigma_{1,l} & & & & & & \\
& & & & \alpha_{1,l} & & & & & \\
& & & & & \beta_{1,l} & & & & \\
& & & & & & \ddots & & & \\
& & & & & & & \sigma_{m,l} & & \\
& & & & & & & & \alpha_{m,l} & \\
& & & & & & & & & \beta_{m,l}
\end{bmatrix}
\tag{76}
$$

$$
\mathbf{S} = \begin{bmatrix}
\mathbf{d}_1^T \\
\vdots \\
\mathbf{d}_n^T \\
\mathbf{e}_1^T + \boldsymbol{f}_1^T \\
\boldsymbol{f}_1^T \\
\mathbf{e}_1^T \\
\vdots \\
\mathbf{e}_m^T + \boldsymbol{f}_m^T \\
\boldsymbol{f}_m^T \\
\mathbf{e}_m^T
\end{bmatrix}
\tag{77}
$$

where $\mathbf{J} \in \mathbb{R}^{S \times (n+3m)}$, $\mathbf{P}_l \in \mathbb{R}^{(n+3m) \times (n+3m)}$, $\mathbf{S} \in \mathbb{R}^{(n+3m) \times S}$, $\alpha_{i,l} = \omega_{i,l} - \sigma_{i,l}$, and $\beta_{i,l} = -\omega_{i,l} - \sigma_{i,l}$, we can diagonalize $(\mathbf{A}(\mathbf{I} - \mathbf{KC}))^l$ as $\mathbf{JP}_l\mathbf{S}$.

Let $\mathbf{V} = (\mathbf{SAK})^T$, $\mathbf{U} = \mathbf{CJ}$, $\mathbf{m} = \mathbf{C}\sum_{l=1}^{L} \mathbf{\Gamma}^l(\mathbf{b} - \mathbf{AKd}) + \mathbf{d}$, and $\mathbf{X}_t = \mathbf{y}_{t-1:t-L}$. Let $\mathbf{W} \in \mathbb{R}^{L \times (n+3m)}$ be a matrix, whose element in the $l^{th}$ row and $k^{th}$ column is $p_{l-1,kk}$ (i.e., the element in the $k^{th}$ row and $k^{th}$ column of $\mathbf{P}_{l-1}$), and $\mathcal{G} \in \mathbb{R}^{(n+3m) \times (n+3m) \times (n+3m)}$ be a diagonal 3-way tensor, where $g_{ijk} = \mathbb{1}_{i=j=k}$. We can then rearrange the mean to:

$$
\mathbf{C}\boldsymbol{\mu}_{t|t-1} + \mathbf{d} \approx \mathbf{C}\sum_{l=1}^{L} \mathbf{E}\mathbf{\Lambda}^{l-1}\mathbf{E}^{-1}\mathbf{AK}\mathbf{y}_{t-l} + \mathbf{C}\sum_{l=1}^{L} \mathbf{\Gamma}^l(\mathbf{b} - \mathbf{AKd}) + \mathbf{d}
\tag{78}
$$

$$
= \mathbf{C}\sum_{l=1}^{L} \mathbf{JP}_{l-1}\mathbf{SAK}\mathbf{y}_{t-l} + \mathbf{m}
\tag{79}
$$

$$
= \mathbf{U}\sum_{l=1}^{L} \mathbf{P}_{l-1}\mathbf{V}^\top\mathbf{y}_{t-l} + \mathbf{m}
\tag{80}
$$

$$
= \sum_{l=1}^{L}\sum_{i}^{n+3m}\sum_{j}^{n+3m}\sum_{k}^{n+3m} g_{ijk}\,\mathbf{u}_{:i} \circ \mathbf{v}_{:j}(p_{l-1,kk}\mathbf{y}_{t-l}) + \mathbf{m}
\tag{81}
$$

$$
= \sum_{i}^{n+3m}\sum_{j}^{n+3m}\sum_{k}^{n+3m} g_{ijk}\,\mathbf{u}_{:i} \circ \mathbf{v}_{:j}(\mathbf{X}_t\mathbf{w}_{:k}) + \mathbf{m}
\tag{82}
$$

$$
= \left[\sum_{i=1}^{n+3m}\sum_{j=1}^{n+3m}\sum_{k=1}^{n+3m} g_{ijk}\,\mathbf{u}_{:i} \circ \mathbf{v}_{:j} \circ \mathbf{w}_{:k}\right] \times_{2,3} \mathbf{X}_t + \mathbf{m}
\tag{83}
$$

And so concludes the proof. $\qquad\square$

## C   Single-subspace SALT

Here we explicitly define the generative model of multi-subspace and single-subspace Tucker-SALT and CP-SALT. Single-subspace SALT is analogous to single-subspace SLDSs (also defined below), where certain emission parameters (e.g., $\mathbf{C}$, $\mathbf{d}$, and $\mathbf{R}$) are shared across discrete states. This reduces the expressivity of the model, but also reduces the number of parameters in the model. Note that both variants of all models have the same structure on the transition dynamics of $\mathbf{z}_t$.

**Multi-subspace SALT**   Note that the SALT model defined in (6) and (7) in the main text is a multi-subspace SALT. We repeat the definition here for ease of comparison.

$$\mathbf{y}_t \overset{\text{i.i.d.}}{\sim} \mathcal{N}\left(\left(\sum_{i=1}^{D_1}\sum_{j=1}^{D_2}\sum_{k=1}^{D_3} g_{ijk}^{(z_t)}\,\mathbf{u}_{:i}^{(z_t)}\circ\mathbf{v}_{:j}^{(z_t)}\circ\mathbf{w}_{:k}^{(z_t)}\right)\times_{2,3}\mathbf{y}_{t-1:t-L}+\mathbf{b}^{(z_t)},\mathbf{\Sigma}^{(z_t)}\right), \tag{84}$$

$D_1 = D_2 = D_3 = D$ and $\mathcal{G}$ is diagonal for CP-SALT.

**Single-subspace Tucker-SALT**   In single-subspace methods, the output factors are shared across discrete states

$$\mathbf{y}_t \overset{\text{i.i.d.}}{\sim} \mathcal{N}\left(\mathbf{U}\left(\mathbf{m}^{(z_t)}+\left(\sum_{j=1}^{D_2}\sum_{k=1}^{D_3}\mathbf{g}_{:jk}^{(z_t)}\circ\mathbf{v}_{:j}^{(z_t)}\circ\mathbf{w}_{:k}^{(z_t)}\right)\times_{2,3}\mathbf{y}_{t-1:t-L}\right)+\mathbf{b},\mathbf{\Sigma}^{(z_t)}\right), \tag{85}$$

where $\mathbf{m}^{(z_t)} \in \mathbb{R}^{D_1}$.

**Single-subspace CP-SALT**   Single-subspace CP-SALT requires an extra tensor compared to Tucker-SALT, as this tensor can no longer be absorbed in to the core tensor.

$$\mathbf{y}_t \overset{\text{i.i.d.}}{\sim} \mathcal{N}\left(\mathbf{U}'\left(\mathbf{m}^{(z_t)}+\mathbf{P}^{(z_t)}\left(\left(\sum_{j=1}^{D_2}\sum_{k=1}^{D_3}\mathbf{g}_{:jk}^{(z_t)}\circ\mathbf{v}_{:j}^{(z_t)}\circ\mathbf{w}_{:k}^{(z_t)}\right)\times_{2,3}\mathbf{y}_{t-1:t-L}\right)\right)+\mathbf{b},\mathbf{\Sigma}^{(z_t)}\right),$$
$$\tag{86}$$

where $\mathbf{U}' \in \mathbb{R}^{N\times D_1'}$, $\mathbf{P}^{(z_t)} \in \mathbb{R}^{D_1'\times D_1}$, $\mathbf{m}^{(z_t)} \in \mathbb{R}^{D_1'}$, $D_1 = D_2 = D_3 = D$, and $\mathcal{G}$ is diagonal.

**Multi-subspace SLDS**   Multi-subspace SLDS is a much harder optimization problem, which we found was often numerically unstable. We therefore do not consider multi-subspace SLDS in these experiments, but include its definition here for completeness

$$\mathbf{x}_t \sim \mathcal{N}\left(\mathbf{A}^{(z_t)}\mathbf{x}_{t-1}+\mathbf{b}^{(z_t)},\mathbf{Q}^{(z_t)}\right), \tag{87}$$

$$\mathbf{y}_t \sim \mathcal{N}\left(\mathbf{C}^{(z_t)}\mathbf{x}_t+\mathbf{d}^{(z_t)},\mathbf{R}^{(z_t)}\right). \tag{88}$$

**Single-subspace SLDS**   Single-subspace SLDS was used in all of our experiments, and is typically used in practice [Petreska et al., 2011, Linderman et al., 2017]

$$\mathbf{x}_t \sim \mathcal{N}\left(\mathbf{A}^{(z_t)}\mathbf{x}_{t-1}+\mathbf{b}^{(z_t)},\mathbf{Q}^{(z_t)}\right), \tag{89}$$

$$\mathbf{y}_t \sim \mathcal{N}\left(\mathbf{C}\mathbf{x}_t+\mathbf{d},\mathbf{R}\right). \tag{90}$$

# D Synthetic Data Experiments

## D.1 Extended Experiments for Proposition 1

In Section 5.1 we showed that Proposition 1 can accurately predict the required rank for CP- and Tucker-SALT models. We showed results for a single LDS for clarity. We now extend this analysis across multiple random LDS and SALT models. We randomly sampled LDSs with latent dimensions ranging from 4 to 10, and observation dimensions ranging from 9 to 20. For each LDS, we fit 5 randomly initialized CP-SALT and Tucker-SALT models with $L = 50$ lags. We varied the rank of our fit SALT models according to the rank predicted by Proposition 1. Specifically, we computed the estimated number of ranks for a given LDS, denoted $D^*$, and then fit SALT models with $\{D^* - 2, D^* - 1, D^*, D^* + 1, D^* + 2\}$ ranks. According to Proposition 1, we would expect to see the reconstruction error of the autoregressive tensor be minimized, and for prediction accuracy to saturate, at $D = D^*$.

To analyze these model fits, we first computed the average mean squared error of the autoregressive tensor corresponding to the LDS simulation, as a function of SALT rank relative to the rank required by Proposition 1. We see, as predicted by Proposition 1, that error in the autoregressive tensor is nearly always minimized at $D^*$ (Figure 8A). Tucker-SALT was always minimized at $D^*$. Some CP-SALT fits have lower MSE at ranks other than predicted by Proposition 1. We believe this is due to local minima in the optimization. We next investigated the test log-likelihood as a function of the relative rank (Figure 8B). Interestingly, the test log-likelihood shows that Tucker-SALT strongly requires the correct number of ranks for accurate prediction, but CP-SALT can often perform well with fewer ranks than predicted (although still a comparable number of ranks to Tucker-SALT). As in Figure 2, these analyses empirically confirm Proposition 1.

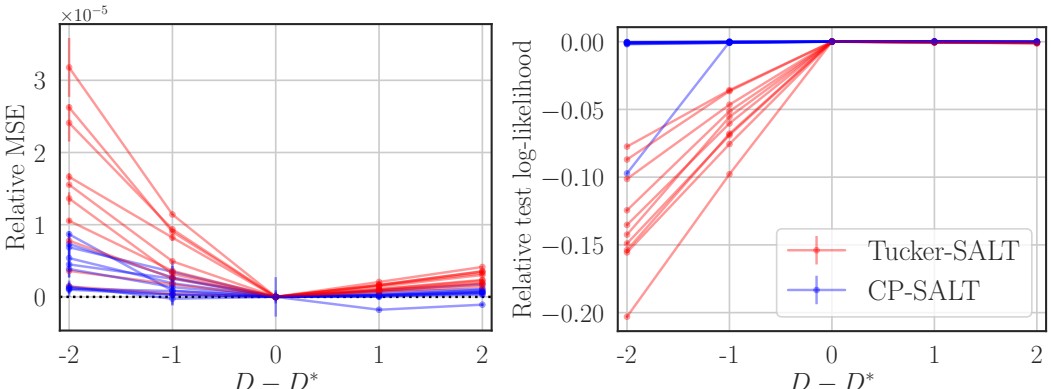

*(a)* Normalized MSE of autoregressive tensor.          *(b)* Normalized log-likelihood on held-out test set.

*Figure 8:* Extended results examining Proposition 1. Results are shown for the ability of SALT to estimate ten randomly generated LDSs, using five SALT repeats for each LDS. MSEs (in panel A) and log-likelihoods (in panel B) are normalized by the mean MSE and mean log-likelihood of SALT models trained with $D = D^*$. $D$ is the rank of the fit SALT model, and $D^*$ is the necessary rank predicted by Proposition 1.

## D.2 Quantitative Performance: Synthetic Switching LDS Experiments

We include further results and analysis for the NASCAR® and Lorenz attractor experiments presented in Section 5.2. We compare the marginal likelihood achieved by single-subspace SALT models of different sizes. We see that SALT outperforms ARHMMs, and can fit larger models (more lags) without overfitting (Figure 9). Note that the SLDS does not admit exact inference, and so we cannot readily compute the exact marginal likelihood for the SLDS.

## D.3 TVART versus SALT in recovering the parameters of SLDSs

We compared SALT to TVART Harris et al. [2021], another tensor-based method for modeling autoregressive processes. We modified TVART (as briefly described in the original paper, Harris et al. [2021]) so that it can handle AR(p) processes, as opposed to only AR(1) processes. TVART is also

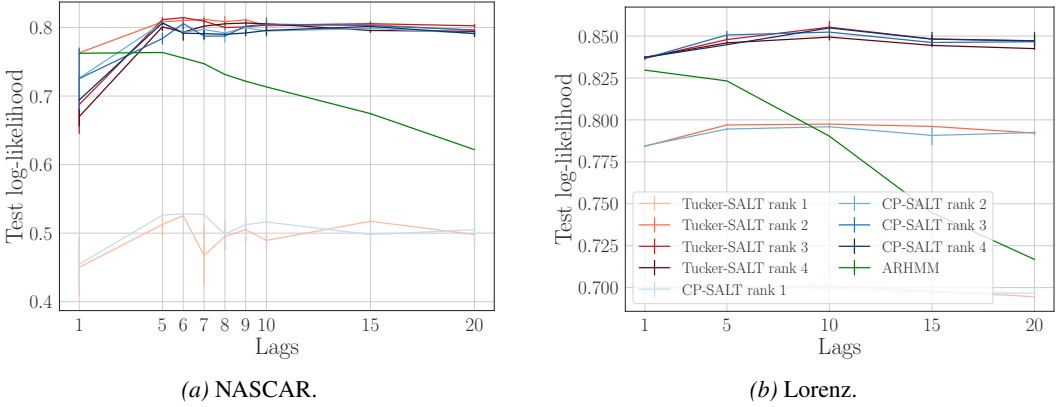

| (a) NASCAR. | (b) Lorenz. |

*Figure 9:* Quantitative performance of different SALT models and ARHMMs (averaged over 3 different runs) on the synthetic experiments presented in Section 5.2. The test-set log likelihood is shown as a function of lags in the SALT model, for both **(A)** the NASCAR® and **(B)** Lorenz synthetic datasets.

not a probabilistic model (i.e., cannot compute log-likelihoods), and so we focus our comparison on how well these methods recover the parameters of a ground-truth SLDS.

We used the same SLDS that we used to generate the NASCAR® dataset in Section 5.2. We then used $L = 7$ CP-SALT and Tucker-SALT with ranks 3 and 2, respectively, and computed the MSE between the ground truth tensor and SALT tensors. For TVART, we used $L = 7$, bin size of 10, and ranks 2 and 3 to fit the model to the data. We then clustered the inferred dynamics parameters to assign discrete states. To get the TVART parameter estimation, we computed the mean of the dynamics parameters for each discrete state and computed the MSE against the ground truth tensor. The MSE results are as follows:

*Table 2:* Results comparing SALT and TVART Harris et al. [2021] on the NASCAR example.

| Model | Rank | Tensor Reconstruction MSE ($\times 10^{-3}$) | Number of parameters |
|---|---|---|---|
| TVART | 2 | 0.423 | 1.4K |
| TVART | 3 | 0.488 | 2.0K |
| Tucker-SALT | 2 | 0.294 | 0.6K |
| CP-SALT | 3 | 0.297 | 0.7K |

Table 2 shows that SALT models recover the dynamics parameters of the ground truth SLDS more accurately. Furthermore, we see that SALT models use fewer parameters than TVART models for the dataset (as the number of parameters in TVART scales linearly with the number of windows). We also note that TVART cannot be applied to held-out data, and, without post-hoc analysis, does not readily have a notion of re-usable dynamics or syllables.

### D.4 The effect of the number of switches on the recovery of the parameters of the autoregressive dynamic tensors

We asked how the frequency of discrete state switches affected SALT's ability to recover the autoregressive tensors. We trained SALT, the ARHMM, all with $L = 5$ lags, and the SLDS on data sampled from an SLDS with varying number of discrete state switches. The ground-truth SLDS model had $H = 2$ discrete states, $N = 20$ observations and $S = 7$ dimensional continuous latent states. The matrix $\mathbf{A}^{(h)}(\mathbf{I} - \mathbf{K}^{(h)}\mathbf{C}^{(h)})$ of each discrete state of the ground-truth SLDS had 1 real eigenvalue and 3 pairs of complex eigenvalues. We sampled 5 batches of $T = 15,000$ timesteps of data from the ground-truth SLDS, with $s_n \in \{1, 10, 25, 75, 125\}$ number of discrete state switches that were evenly spaced out across the data. We then computed the mean squared error (MSE) between the SLDS tensors and the tensors reconstructed by SALT, the ARHMM, and the SLDS. (Figure 10). More precisely, we combined the 3rd order autoregressive tensors from each discrete state into a 4th order tensor, and calculated the MSE based on these 4th order tensors. As expected, the MSE

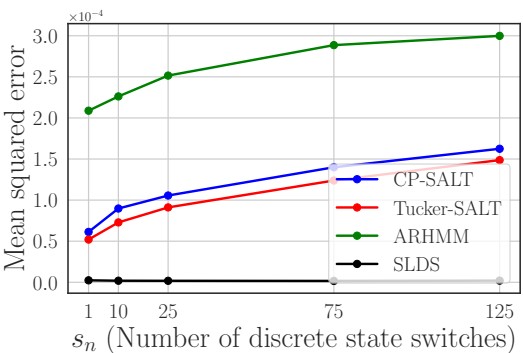

*Figure 10:* **The quality of SALT approximation of SLDSs decreases as the number of discrete state switches increases:** The data comes from an SLDS with $H = 2$, $N = 20$, and $S = 7$. 15,000 timesteps were generated, with varying numbers of evenly spaced discrete state switches (x-axis). The mean squared error of reconstructing the autoregressive tensors increased as a function of the number of discrete state switches. Note that we combined the 3rd order autoregressive tensors from each discrete state into a 4th order tensor, and calculated the MSE based on these 4th order tensors.

increased with the number of switches in the data, indicating that the quality of SALT approximation of SLDSs decreases as the frequency of discrete state switches increases.

# E Modeling Mouse Behavior

We include further details for the mouse experiments in Section 5.3.

## E.1 Training Details

We used the first 35,949 timesteps of data from each of the three mice, which were collected at 30Hz resolution. We used $H = 50$ discrete states and fitted ARHMMs and CP-SALT models with varying lags and ranks. Similar to Wiltschko et al. [2015], we imposed stickiness on the discrete state transition matrix via a Dirichlet prior with concentration of 1.1 on non-diagonals and $6 \times 10^4$ on the diagonals. These prior hyperparameters were empirically chosen such that the durations of the inferred discrete states and the given labels were comparable. We trained each model 5 times with random initialization for each hyperparameter, using 100 iterations of EM on a single NVIDIA Tesla P100 GPU.

## E.2 Video Generation

Here we describe how the mouse behavioral videos were generated. We first determined the CP-SALT hyperparameters as those which led to the highest log-likelihood on the validation dataset. Then, using that CP-SALT model, we computed the most likely discrete states on the train and test data. Given a discrete state $h$, we extracted slices of the data whose most likely discrete state was $h$. We padded the data by 30 frames (i.e. 1 second) both at the beginning and the end of each slice for the movie. A red dot appears on each mouse for the duration of discrete state $h$. We generated such videos for all 50 discrete states (as long as there existed at least one slice for each discrete state) on the train and test data. For a given discrete state, the mice in each video behaved very similarly (e.g., the mice in the video for state 18 "pause" when the red dots appear, and those in the video for state 32 "walk" forward), suggesting that CP-SALT is capable of segmenting the data into useful behavioral syllables. See "MoSeq_salt_videos_train" and "MoSeq_salt_videos_test" in the supplementary material for the videos generated from the train and test data, respectively. "salt_crowd_$i$.mp4" refers to the crowd video for state $i$. We show the principal components for states $1, 2, 13, 32, 33, 47$ in Figure 11.

## E.3 Modeling Mouse Behavior: Additional Analyses

We also investigated whether SALT qualitatively led to a good segmentation of the behavioral data into discrete states, shown in Figure 11. Figure 11A shows a 30 second example snippet of the test data from one mouse colored by the discrete states inferred by CP-SALT. CP-SALT used fewer discrete states to model the data than the ARHMM (Figure 11B). Coupled with the finding that CP-SALT improves test-set likelihoods, this suggests that the ARHMM may have oversegmented the data and CP-SALT may be better able to capture the number of behavioral syllables. Figure 11C shows average test data (with two standard deviations) for a short time window around the onset of a discrete state (we also include mouse videos corresponding to that state in the supplementary materials). The shrinking gray area around the time of state onset, along with the similar behaviors of the mice in the video, suggests that CP-SALT is capable of segmenting the data into consistent behavioral syllables.

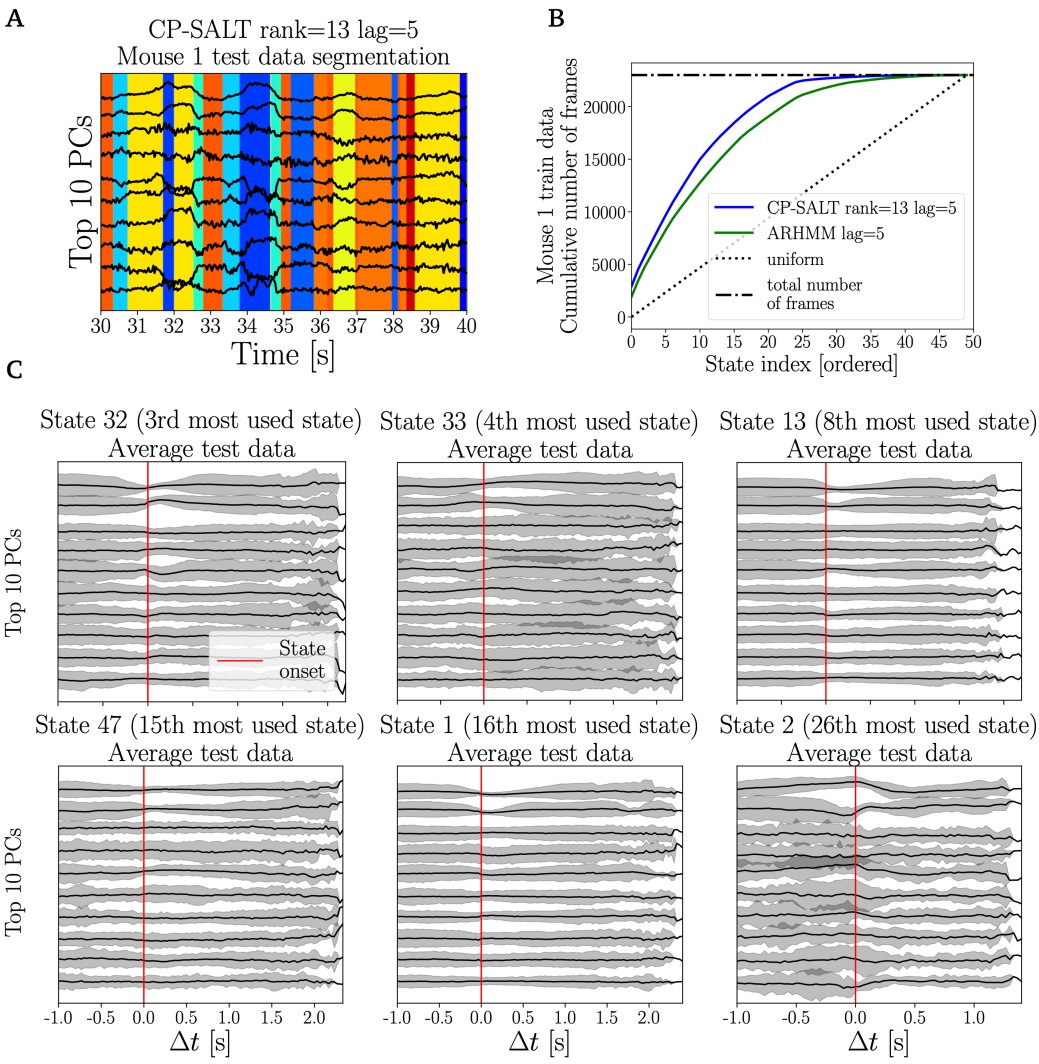

*Figure 11:* **CP-SALT leads to qualitatively good segmentation of the mouse behavioral data into distinct syllables.**: **(A)** 30 seconds of test data (Mouse 1) with the discrete states inferred by CP-SALT as the background color. **(B)** For one mouse, the cumulative number of frames that are captured by each discrete state, where the discrete states are ordered according to how frequently they occur. **(C)** The average test data, with two standard deviations, for six states of CP-SALT, aligned to the time of state onset. The shrinkage of the gray region around the state onset tells us that CP-SALT segments the test data consistently.

# F Modeling *C. elegans* Neural Data

We include further details and results for the *C. elegans* example presented in Section 5.4. This example highlights how SALT can be used to gain scientific insight in to the system.

## F.1 Training Details

We used ∼3200 timesteps of data (recorded at 3Hz) from one worm, for which 48 neurons were confidently identified. The data were manually segmented in to seven labels (reverse sustained, slow, forward, ventral turn, dorsal turn, reversal (type 1) and reversal (type 2). We therefore used $H = 7$ discrete states in all models (apart from the GLM). After testing multiple lag values, we selected $L = 9$ for all models, as these longer lags allow us to examine longer-timescale interactions and produced better segmentations across models, with only a small reduction in variance explained. We trained each model 5 times with KMeans initialization, using 100 iterations of EM on a single NVIDIA Tesla V100 GPU. Models that achieved 90% explained variance on a held-out test set were then selected and analyzed (similar to Linderman et al. [2019]).

## F.2 Additional Quantitative Results

Figure 12 shows additional results for training different models. In Figure 12A we see that models with larger ranks (or latent dimension) achieve higher explained variance. Interestingly, longer lags can lead to a slight reduction in the explained variance, likely due to overfitting. This effect is less pronounced in the more constrained single-subspace SALT, but, these models achieve lower explained variance ratios throughout. Longer lag models allow us to inspect longer-timescale dependencies, and so are more experimentally insightful. Figure 12B shows the confusion matrix for discrete states between learned models and the given labels. The segmentations were similar across all models that achieved 90% explained variance.

## F.3 Additional Autoregressive Filters

Figures 13 and 14 show extended versions of the autoregressive filters included in Section 5.4. Figure 13 shows the filters learned for ventral and dorsal turns (for which panel A was included in Figure

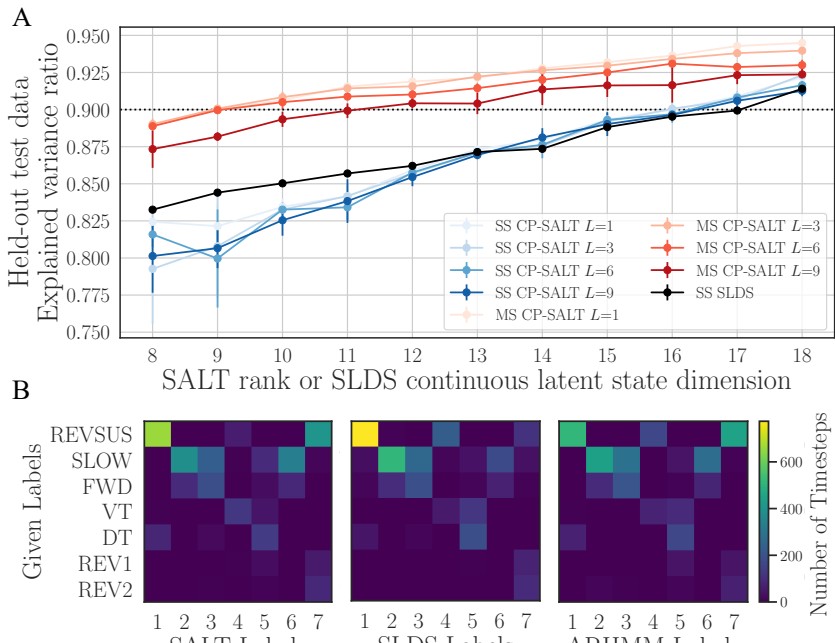

*Figure 12:* **SALT and SLDS perform comparably on held-out data**: **(A)**: Explained variance on a held-out sequence. Single-subspace (SS) SALT and SLDS perform comparably. Multi-subspace (MS) SALT achieves a higher explained variance with fewer ranks. Multi-subspace SLDS was numerically unstable. **(B)**: Confusion matrices between given labels and predicted labels. All methods produce similar quality segmentations.

5), while Figure 14 shows the filters for forward and backward locomotion. Note that the GLM does not have multiple discrete states, and hence the same filters are used across states. We see for ARHMM and SALT that known-behavior tuned neurons have higher magnitude filters (determined by area under curve), whereas the SLDS and GLM do not recover such strong state-specific tuning. Since the learned SLDS did not have stable within-state dynamics, the autoregressive filters could not be computed using Equation (48). We thus show $CA^{(h)l}C^+$ for lag $l$, where $C^+$ denotes the Moore-Penrose pseudoinverse of $C$, as a proxy for the autoregressive filters of discrete state $h$ of the SLDS. Note that this is a post-hoc method and does not capture the true dependencies in the observation space.

We see that SALT consistently assigns high autoregressive weight to neurons known to be involved in certain behaviors (see Figures 13 and 14). In contrast, the ARHMM identifies these relationships less reliably, and the estimate of the SLDS autoregressive filters identifies few strong relationships. As the GLM only have one "state", the autoregressive filters are averaged across state, and so few strong relationships are found. This highlights how the low-rank and switching properties of SALT can be leveraged to glean insight into the system.

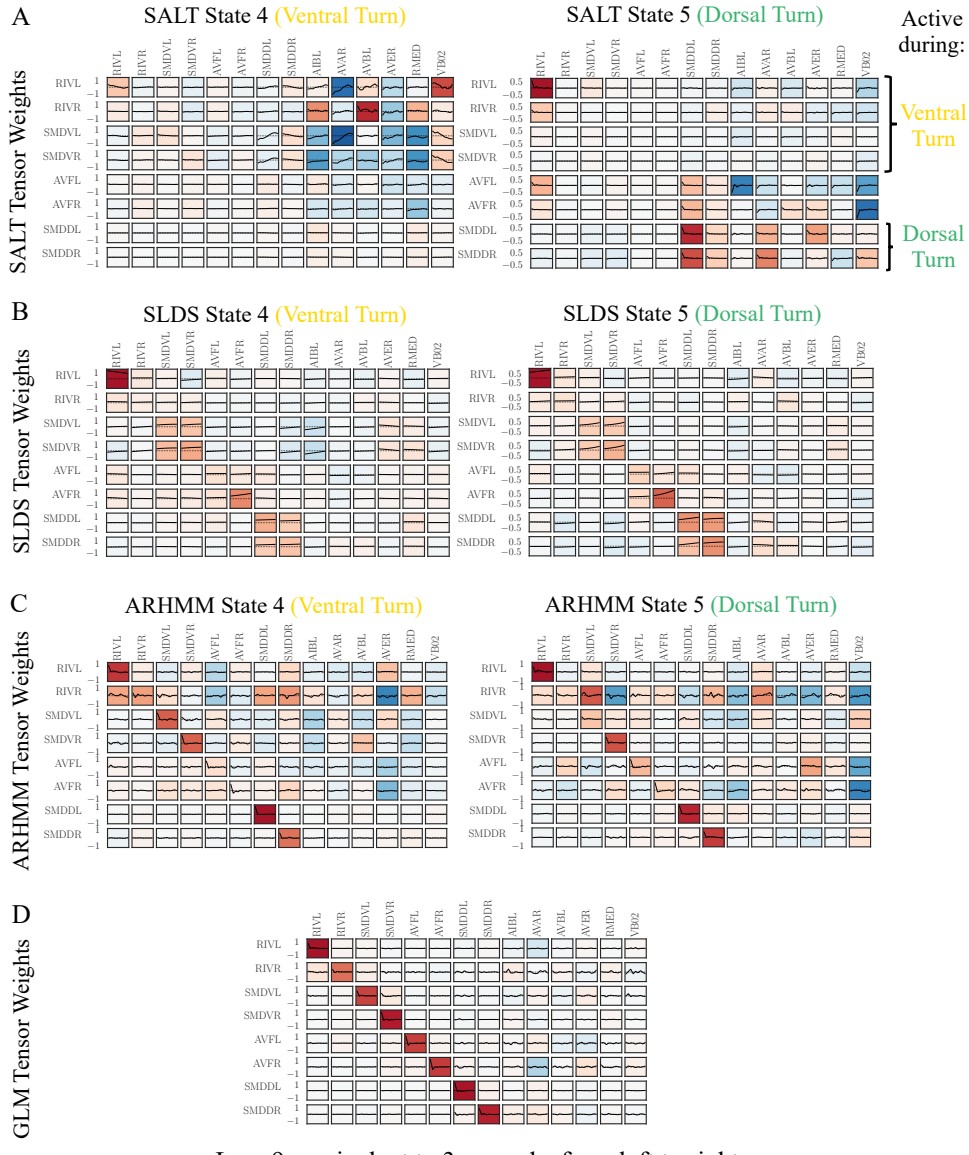

Lag=9, equivalent to 3 seconds; from left to right

*Figure 13:* **Autoregressive tensors learned by different models (Ventral and Dorsal Turns): (A-C)** One-dimensional autoregressive filters learned in two states by SALT, SLDS, ARHMM (identified as ventral and dorsal turns), and **(D)** by a GLM. `RIV` and `SMDV` are known to mediate ventral turns, while `SMDD` mediate dorsal turns [Kato et al., 2015, Gray et al., 2005, Yeon et al., 2018].

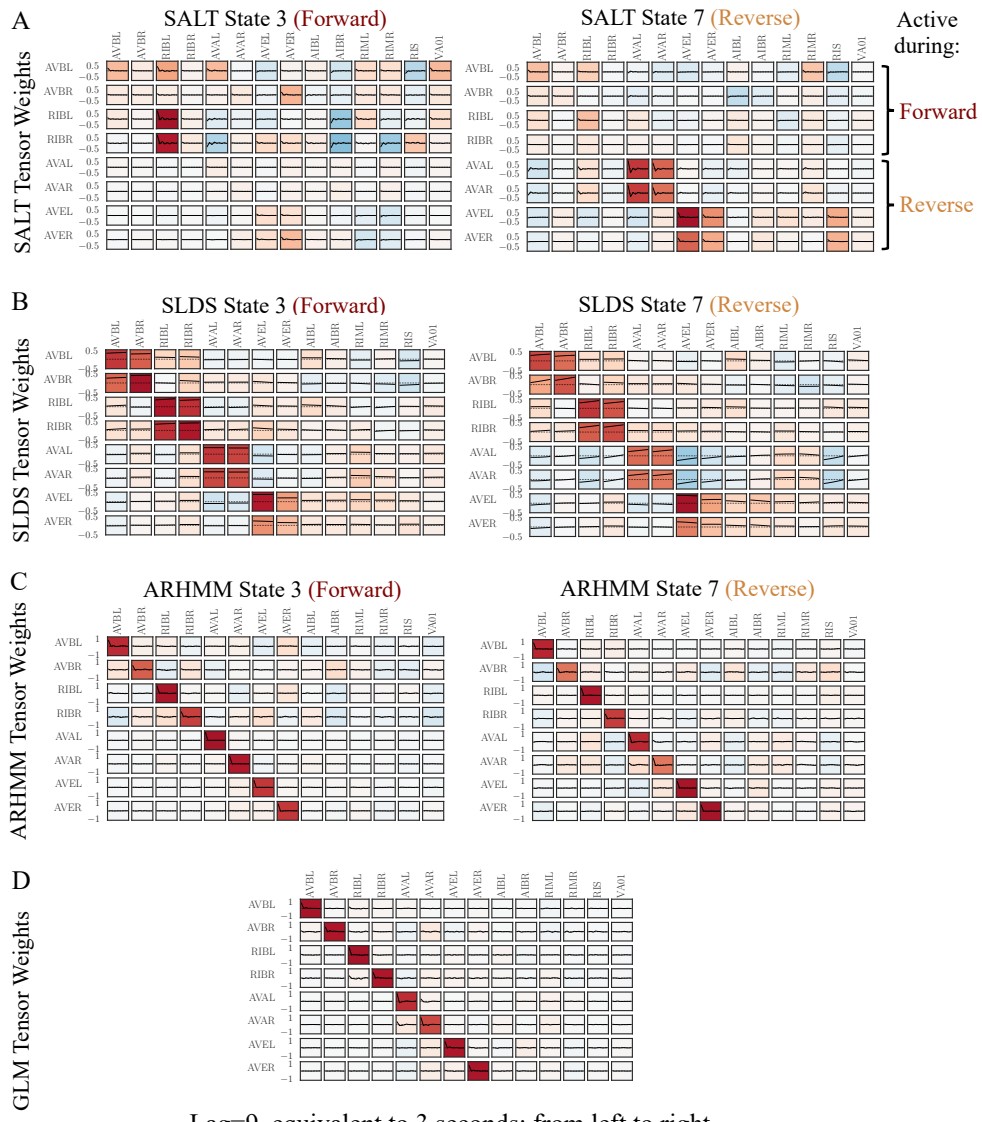

Lag=9, equivalent to 3 seconds; from left to right

*Figure 14:* **Autoregressive tensors learned by different models (Forward Locomotion and Reversal)**: **(A-C)** One-dimensional autoregressive filters learned in two states by SALT, SLDS, ARHMM (identified as forward and reverse), and **(D)** by a GLM. AVB and RIB are known to mediate forward locomotion, while AVA and AVE are involved in initiating reversals [Kato et al., 2015, Gray et al., 2005, Chalfie et al., 1985, Piggott et al., 2011].

## G Deep Switching Autoregressive Model

Here we compare SALT to *deep switching auto-regressive factorization* (DSARF) models [Farnoosh et al., 2021]. DSARF models construct a deep generative model using a set of low-rank factors, which are then weighted according to inferred discrete states at each timestep. This combination then defines the time evolution of an autoregressive process, which in turn defines the distribution over the observed variables. As such, non-linear function approximators can be used to parameterize many of the link functions, gaining expressivity, but retaining the parameter efficiency and interpretability of conventional methods. Variational inference is used to learn the parameters and perform inference. Unlike SALT, DSARF *can* handle missing data, as autoregressive processes are defined only in the latent space.

We compare SALT against DSARF on an example drawn from Farnoosh et al. [2021], proposed for studying switching systems by Ghahramani and Hinton [2000] (see also Weigend and Gershenfeld [1994]). This example studies a patient believed to have sleep apnea, typified by periods where normal rhythmic breathing ceases, resulting in periods of low or zero respiratory rates. The data are a one-dimensional time series of a measure of chest volume, such that oscillations in the data correspond to rhythmic breathing (see Figure 15). Periods of constant volume correspond to apnea bouts. Disjoint one thousand length sequences are used for train and test sets, such that $\mathbf{y}^{\text{train}} \in \mathbb{R}^{1 \times 1000}$ and $\mathbf{y}^{\text{test}} \in \mathbb{R}^{1 \times 1000}$.

We apply DSARF as described in Farnoosh et al. [2021] and SALT. The SALT model we used uses $H = 2$ discrete states, $D = 5$ ranks and $L = 10$ lags, and with a single latent subspace. We use an L2 weight penalty of $10^{-4}$. As per the mouse experiments, we add a Dirichlet stickiness prior with parameters $\gamma = 10^{-2}$ and $\kappa = 10^3$ (See Table 3). Inference results on the test set are shown in Figure 15. SALT hyperparameters were selected through manual tuning. SALT achieves a normalized next-step prediction RMSE of 22.57%, vs 23.86% achieved by DSARF (described by [Farnoosh et al., 2021] as "short-term prediction") This result highlights that SALT is competitive with "deep" methods that provide the desired discrete-continuous representation.

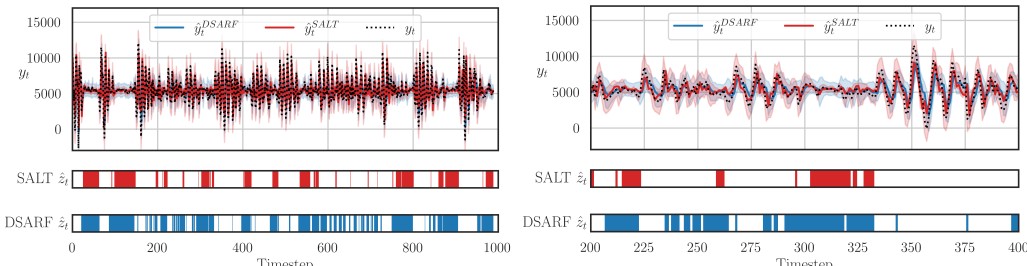

*Figure 15:* Comparison of DSARF and SALT-CP on the apnea example as presented in Farnoosh et al. [2021] Shown are *filtering* reconstructions of the observed trace and binary discrete label. SALT achieves a normalized RMSE of 22.57% vs 23.86% achieved by DSARF. Right panel is a zoomed in version of the left panel. We see that reconstructions and scores achieved by both models are comparable.

# H  Experiment Configurations

In this section we provide extended details for the experiments presented in the main text, including the hyperparameters selected and the hyperparameter tuning process.

## H.1  List of Hyperparameters

Table 3 outlines the key hyperparameters we consider when comparing and selecting models. Details of the values for each of these hyperparameters are then specified in the sections afterwards.

*Table 3:* Outline of the key hyperparameters in the three main models families we consider: SALT, SLDS and ARHMM.

| Hyperparameter | Description | Applicable to models | Permissible Values |
|---|---|---|---|
| Tensor factorization | Factorization structure of autoregressive tensor | SALT, ARHMM | {CP, Tucker, None} |
| Subspace type | Whether the output factors are shared across discrete states. | SALT, SLDS | {Single, Multi} |
| $D$, Tensor ranks | Dimension of the core tensor. Assumed throughout that $D = D_1 = D_2 = D_3$. | SALT | $\mathbb{Z}_{\geq 1}$ |
| $H$, Number of discrete states | Number of discrete states in switching models. | SALT, SLDS, ARHMM | $\mathbb{Z}_{\geq 1}$ |
| $L$, Number of lags | Number of previous observations in autoregressive models. | SALT, ARHMM | $\mathbb{Z}_{\geq 1}$ |
| $S$, Latent space dimension | Dimension of continuous latent state. | SLDS | $\mathbb{Z}_{\geq 1}$ |
| Temporal L2 penalty | L2 penalty applied lag parameters at longer lags. The penalty strength is defined as $\alpha \cdot \beta^{l-1}$ where $l$ is the lag. | SALT, ARHMM | $\alpha \in \mathbb{R}_{\geq 0},\ \beta \in \mathbb{R}_{\geq 1}$ |
| Stickiness | Dirichlet prior that can be added to penalize discrete state switches. | SALT, SLDS, ARHMM | $\gamma \in \mathbb{R}_{>0}, \kappa \in \mathbb{R}_{\geq 0}$ |
| Global L2 penalty | L2 penalty applied across all parameters. (See note below) | SALT, SLDS, ARHMM | $\mathbb{R}_{\geq 0}$ |

Unless otherwise specified, we performed a grid search over a range of values within the permissible set. In certain circumstances, the hyperparameter was selected to match known properties of the data, e.g. we used $H = 4$ for the NASCAR data, because we know there are four underlying states in the data.

During pilot experiments we experimented with a global L2 penalty applied to all parameters in the model. We found that varying this parameter did not affect the key outcomes of each experiment. We therefore set the L2 penalty strength to the same value for all the models for all the experiments, with strength $0.0001$, unless otherwise specified.

## H.2  Experiment Specific Hyperparameters

Here we specify experiment-specific details.

### H.2.1  Section 5.1: SALT Faithfully Approximates LDS

For a given LDS with latent dimension size of $S$ and initialized with a random rotation matrix, we computed the estimated number of CP-SALT and Tucker-SALT ranks, denoted $D^*_{\text{CP}} = n + 3m$ and $D^*_{\text{Tucker}} = n + 2m$, respectively, where $n$ is the number of real eigenvalues and $m$ is the number of complex conjugate pairs of $\Gamma$ defined in Proposition 1. We then fitted CP-SALT models with $\{\min(S, D^*_{\text{CP}}) - 2, \dots, \max(S, D^*_{\text{CP}}) + 2\}$ and Tucker-SALT models with $\{\min(S, D^*_{\text{Tucker}}) - 2, \dots, \max(S, D^*_{\text{Tucker}}) + 2\}$. We chose $L = 50$, as this was the horizon at which the autoregressive parameters were approximately zero. The temporal L2 penalty was set to $1.0$.

### H.2.2  Section 5.2: Synthetic Switching LDS Examples

**Synthetic NASCAR dataset**    All models used the true number of discrete latent states, $H = 4$. We fitted single-subspace CP-SALT and Tucker-SALT models with ranks $D \in \{1, 2, 3, 4\}$. For both SALT models and ARHMMs, we used $L \in \{1, 5, 6, 7, 8, 9, 10, 15, 20\}$. A temporal L2 penalty of $1.0$ was used for CP-SALT and Tucker-SALT models. Single-subspace SLDSs were fitted with the true underlying latent dimension, $S = 2$.

**Synthetic Lorenz attractor dataset**    All models used the approximate number of discrete latent states in the data, $H = 2$ (following Linderman et al. [2017]). We fitted single-subspace CP-SALT and Tucker-SALT models with ranks $D \in \{1, 2, 3, 4\}$. For both SALT models and ARHMMs, we used $L \in \{1, 5, 10, 15, 20\}$. A temporal L2 penalty of $2.0$ was used for CP-SALT and Tucker-SALT models. Single-subspace SLDSs were fitted with the true underlying latent dimension, $S = 3$.

### H.2.3  Section 5.3: Modeling Mouse Behavior

We fitted multi-subspace CP-SALT models with ranks $D \in \{10, 11, 12, 13, 14\}$. For both CP-SALT models and ARHMMs, we used $L \in \{1, 2, 3, 4, 5, 6, 7, 8, 9, 10, 15, 20\}$ and the number of discrete states was set to $H = 50$ (51 behavioral states explained 95% of videos Wiltschko et al. [2015]). We use a temporal L2 penalty of 1.0. Similar to Wiltschko et al. [2015], we imposed stickiness on the discrete state transition matrix of both SALT models and ARHMMs via a Dirichlet prior. For discrete state $h$, the concentration parameters $\nu \in \mathbb{R}^H_{>0}$ of the Dirichlet prior is $\gamma$ for $\nu_i$, $i \neq h$, and $\gamma + \kappa$ for $\nu_h$. For this experiment, $\gamma$ was set to 1.1 and $\kappa$ to $6 \times 10^4$, which were empirically chosen such that the duration of the inferred discrete states and the given labels were comparable.

### H.2.4  Section 5.4: Modeling *C. elegans* Neural Data

Following Linderman et al. [2019], we empirically searched for sets of hyperparameters that achieve $\sim$90% explained variance on a held-out test dataset. We fitted both single and multi-subspace CP-SALT models with ranks $D \in \{8, 9, 10, 11, 12, 13, 14, 15, 16, 17, 18\}$. Similarly, SLDSs were fitted with the same range of latent dimension size. For both CP-SALT models and ARHMMs, we used $L \in \{1, 3, 6, 9\}$ and the number of discrete states was set to $H = 7$, which is the number of unique manual labels. After testing multiple lag values, we selected $L = 9$ for SALT models and ARHMMs, as these longer lags allow us to examine longer-timescale interactions and produced better segmentation across models, with only a small reduction in variance explained. For ARHMMs, we set the global L2 penalty strength to 20.0 with the temporal L2 penalty set to 1.0. For SLDSs, we set the global L2 penalty strength to 100.0. For SALT models, we set the global L2 penalty strength to 0.2 and the temporal L2 penalty to 1.1. We additionally imposed a smoothness penalty to the weights of the lag factors of SALT models by adding $\eta||\mathbf{FW}||_2^2$ to the NLL for optimizing the lag factors, where $\eta$ is the smoothness penalty strength and $\mathbf{F}$ is the first difference matrix. We set $\eta = 0.8$ for SALT models.

