# OpenReview forum: "Switching Autoregressive Low-rank Tensor Models"
_NeurIPS.cc/2023/Conference — NeurIPS 2023 poster_

### Official Review · Reviewer_4hZ1 · 2023-06-19

**Soundness:** 4 excellent
**Presentation:** 4 excellent
**Contribution:** 3 good
**Rating:** 8
**Confidence:** 3

**Summary:**

This paper introduces a new unsupervised probabilistic model for time series analysis: the Switching Autoregressive Low-rank Tensor (SALT). It combines a Low Rank Tensor parametrization of autoregressive (AR) models with switching Dynamics. The two main contributions are:

1)  The SALT model itself. Although the tensor parametrization is not new, they succesfully augment it with switching dynamics and derive an EM-based learning and inference algorithm with closed form updates.

2) A theoretical connection between Linear Dynamical Systems (LDS) and Low-Rank Tensor Autoregression.

The model is evaluated on simulated, behavioral and neural datasets.

**Strengths:**

The paper is very well written despite the need for numerous notations. The author fluently expose their model and relate it to existing approaches.

The link between LDS and Low-Rank Tensor AR models is important and novel (to my understanding). Not only it helps grasping the relationship between those two models, but it also provides a bound linking the dynamics spectrum, the tensor rank and the approximation error one makes when using the tensor AR model. Among other things, I believe it can guide the specification of the tensor rank, which is a non trivial operation.

SALT builds upon existing approaches, and although it is not more expressive than sLDS, inference and learning use closed form updates which is very convincing from a practical perspective. Importantly, it retains the possibility to analyze a low dimensional representation of the observed time series.

Both simulated and real world experiments are convincing and the code is provided.

**Weaknesses:**

1) Title 3.3 is confusing. I agree that the theoretical connection discovered by the author is significant, but it does not concern SALT and sLDS. The link is between LDS and Low Rank Tensor AR models.

2) I think the graphical of Figure 1 is wrong. Why isn't there any arrow between $z_t$ and $z_{t+1}$ despite $z_{t+1} \sim \text{Cat}(\pi^{(z_t)})$ ?


**Questions:**

1) Would it be possible to concatenate the 3-way autoregressive tensors $\mathcal{A}^{h}$ in a 4-way low rank tensor $\mathcal{A}$ of size $N \times N \times L \times h$ ? This might yield an even more compact description of the data.

2) I understood Proposition 1 was a novel contribution from the authors. If it's indeed the case, it's worth mentioning it (more) explicitly in the introduction.

3) How stable are the CP-SALT discovered factors from one initialization to the next ? It could be interesting to look at the factors directly to characterize dynamics, but I think the discovered dynamics could be roughly preserved despite different $\mathcal{A}^{h}$ ?

**Limitations:**

1) The proposed method is based on Gaussian noise models, which is acknowledged by the author. In its current form, it cannot, for example, efficiently handle image data without pre-processing steps.

2) The link between SALT and SLDS is "only" demonstrated for non switching dynamics ($h=1$ ).

---

> ### Author Rebuttal · Authors · 2023-08-10
>
> # Response to Reviewer 4hZ1
>
> We thank the reviewer for the time to review our submission; for their detailed and insightful review; and for highlighting the contributions and the clarity of our submission.
>
> ## R.E. Weaknesses:
>
> **1. Section title:**
>
> >Title 3.3 is confusing. I agree that the theoretical connection discovered by the author is significant, but it does not concern SALT and sLDS. The link is between LDS and Low Rank Tensor AR models.
>
> We have removed switching from the section title.
>
> **2. Graphical model:**
> >I think the graphical of Figure 1 is wrong. Why isn't there any arrow between z_t and z_t+1?
>
> Great catch – we also noticed this the day after submission…  Yes, there should be a link between z’s.  We have corrected the figure.
>
> ## R.E. Questions:
>
> **1.  Concatenating into 4-way tensor:**
>
> >Would it be possible to concatenate the 3-way autoregressive tensors A^h in a 4-way low rank tensor A of size N x N x L x H? This might yield an even more compact description of the data.
>
> Concatenating across states is certainly possible.  However, it isn’t immediately clear how to “mine” additional understanding from the representation.
>
> **2.  Highlight novelty of Proposition 1:**
>
> >I understood Proposition 1 was a novel contribution from the authors. If it's indeed the case, it's worth mentioning it (more) explicitly in the introduction.
>
> Thank you, yes, Proposition 1 is novel.  We have highlighted this more clearly in the introduction.
>
> **3.  Stability of factors:**
>
> >How stable are the CP-SALT discovered factors from one initialization to the next ? It could be interesting to look at the factors directly to characterize dynamics, but I think the discovered dynamics could be roughly preserved despite different A^h?
>
> For the NASCAR and Lorenz experiments, the factors are very stable (up to permutations and rotations) between runs.  For the worm and mice experiments, they are slightly more variable.  We have added a brief discussion of this, explicitly highlighting the equivalence classes and how these can be examined/identified.
>
> ## R.E. Limitations:
>
> 1.  **Handling alternative modalities:** See General Response.  SALT could be easily applied to handle, for instance, image embeddings generated by a deep autoencoder.  This is a very interesting direction for future research!
>
> 2.  **Links between SALT and SLDS around switches:**  We attempted to analyze both SALT and SLDS around switches.  However, the analysis rapidly became intractable.  Practically speaking however, we only require that SALT models produce “similar” switching behavior.  Further understanding the theoretical difference between the two methods is an interesting and challenging opportunity for follow-up work.
>
>
> **Thank you again for your response.**  If you have further questions, we are happy to answer them!
>
> --- The SALT authors.

---

> > ### Comment · Reviewer_4hZ1 · 2023-08-11
> >
> > I thank the authors for their clarification and additional analysis. I maintain my strong accept score of 8.

---

### Official Review · Reviewer_6MVR · 2023-06-30

**Soundness:** 3 good
**Presentation:** 3 good
**Contribution:** 3 good
**Rating:** 7
**Confidence:** 4

**Summary:**

The paper introduces a new model for time-series called SALT (switching autoreg. low-rank tensor). The goal with SALT is to offer a "best of both worlds" alternative to AR-HMMs and switching linear dynamical systems (SLDS).

SALT's relative advantages are:
* enjoys closed-form parameter estimation (unlike SLDS)
* enjoys lower parameter count, meaning less likely to overfit for long-range models on small datasets (unlike AR-HMM)

|                  | AR-HMM      | SALT        | SLDS                  |
|------------------|-------------|-------------|-----------------------|
| param estimation | closed-form EM | closed-form EM | need MCMC/variational |
| parameter count  | O(N^2)      | O(N)        | O(N)                  |
| hyperparameters  | H, L        | D, H, L     | D, L                  |

where

D = rank / number of latent state dims,
H = num hidden states,
L = autoregressive lag

The key idea behind SALT is to essentially force the autoregressive coefficient matrix of an AR-HMM to take a specific factorization structure (See Eq 6-7).

In Sec 3.2, a closed-form EM estimation algorithm is derived.

In Sec 3.3, Prop 1 presents a formal error analysis of how well a SALT model can approximate a *stable* LDS

Experiments in Sec 4 cover

* 5.1: analysis of Prop 1 ideas on a simulated dataset
* 5.2: evidence that SALT might be as expressive as SLDS on synthetic data
* 5.3/5.4: real experiments on mouse behavior and worm neurons


**Strengths:**


+ elegant proposal for factorizing coefficients of AR-HMM
+ parameter estimation via EM allows all steps to be closed-form
+ thoughtful experiments on synthetic and real data

**Weaknesses:**


### Why prefer SALT over SLDS? What real computational advantage?

In Sec 5.4, on a big real dataset it is argued that SALT "can perform as well as" SLDS. While this is all fine, is there a strong advantage that makes a practioner studying this data prefer SALT?

Seems like ultimately, parameter count SALT is larger than SLDS (Table 1).

Closed-form parameter estimation is nice of course, but if the approximate methods for SLDS worked well enough on this dataset, when should a practitioner think that SALT is preferrable, if ever?


### Missing comparison to AR-HMM with L2-regularized coefs

The paper's story suggests that AR-HMM overfitting is the major reason a new model is needed. However, a natural way to prevent overfitting (that preserves closed-form EM parameter estimation) is to provide L2-regularization of (some) AR coefficients. As best as I can tell, the results shown emphasize maximum likelihood estimation, without regularization.

While SALT is elegant, why should a practitioner invest in SALT and its more complex parameterization over a well-studied way to prevent overfitting?

### Hyperparameter settings need to be clarified

**Update after response**: The author response resolved this concern. Text below from original maintained for posterity.

In my understanding, SALT needs the user to select D, H, and L as hyperparameters. In the main paper, I found it difficult to understand which settings were used in several cases and how those were determined.

My concern is that their SALT model could be using substantially higher D/H/L values than alternatives. There needs to be a transparent way these hyperparameters are selected on each dataset.

I'd suggest in Fig 3 caption, reporting N/D/H/L for both racetrack and Lorenz data. Similarly with Fig 4 for mouse data and Fig 5 for worm neural activity.

### Missing some related work

**Update after response**: The author response resolved this concern. Text below from original maintained for posterity.

I'd suggest the authors consider a conceptual comparison to deep switching auto-regressive factorization (DSARF), published by Farnoosh et al at AAAI '21.

Like SALT, DSARF produces discrete segmentations and continuous latent trajectories for a time-series. SALT's factorization is conceptually distinct and has closed-form estimation, while DSARF requires non-conjugate inference via VI with Monte Carlo approximations of gradients. However, DSARF could be more flexible.

**Questions:**


1. How can we use Prop. 1 to decide which D value is optimal? Caption of Fig 2 implies we can use Prop. 1 to pick/rank D for SALT given a known LDS model. But doesn't Prop 1 as stated just provide a statement on error when SALT's D is set to one specific value D = n + 2m?

2. Seems like SALT has more hyperparameters (H, L, and D) to select than either AR-HMM or SLDS (each requires 2 of the 3). Are there good rules of thumb to avoid the expense of 3-dimensional grid search?

3. Can you report typical runtimes on the large datasets you study? Not just for training, but also hyperparameter selection as well.



**Limitations:**

Sec. 6 is missing entirely a proper paragraph discussing limitations of the work at the end of the paper.

---

> ### Author Rebuttal · Authors · 2023-08-10
>
> # Response to Reviewer 6MVR:
>
> We thank the reviewer for taking the time to read our submissions and for their detailed and insightful feedback.  We especially appreciated the description of our method as “elegant”!
>
> The main theme of your review seems to be centred on the experimental utility of SALT – in terms of comparative performance compared and hyperparameter tuning.   We provided some feedback in our General Response, and provide more detailed responses here.
>
> ## R.E. Weaknesses:
>
> **1.  Why prefer SALT over SLDS? What real computational advantage?**
>
> >In Sec 5.4, on a big real dataset it is argued that SALT "can perform as well as" SLDS. While this is all fine, is there a strong advantage that makes a practitioner studying this data prefer SALT?  [...]
>
> A key advantage of SALT over SLDS is that exact inference (posterior over state and evaluation of the log marginal likelihood) in SALT models is straightforward; whereas SLDS can only evaluate a bound on the likelihood and a variational posterior over the latent state (or resort to MCMC).  This makes model development more straightforward and model comparison more accurate.  Inference in SALT models is notably faster than (even approximate) inference in SLDS models.  Finally, the autoregressive dependency structure parameterised by SALT allows more direct insight into the system, compared to the somewhat abstract latent states recovered by SLDS.  The factors can be individually inspected to garner further insight into the system.
>
> *We stress, however, that SALT analysis is a good complement to conventional SLDS analysis, providing different, but equally useful, insights into the system.*
>
> **2. Missing comparison to AR-HMM with L2-regularized coefs:**
>
> >The paper's story suggests that AR-HMM overfitting is the major reason a new model is needed. [...]
>
> We refer the reviewer to General Response B for general discussion of the practical utility of SALT.  Beyond this, however, SALT recovers a low-dimensional continuous description of the data as well, defined as the vector value computed prior to multiplication with the output factors.  The example traces of this low-dimensional variable are shown in Figure 3.  We have highlighted this benefit over ARHMMs in the updated text.
>
> As noted in General Response B, there is no consensus on the best or most principled way to regularize the tensor of an ARHMM, and our experiences in previous work was one of the motivations behind SALT!  For fair comparisons across all models, we opted for the simplest approach and used L2 regularization.  We have added further discussion of this point to the main text.
>
>
> **3. Hyperparameter settings need to be clarified:**
>
>
> >In my understanding, SALT needs the user to select D, H, and L as hyperparameters.  [...]
>
> The parameter complexities of SALT and SLDS were comparable across most of the models we considered.  Note that for the C. elegans experiment, we used longer lags than was strictly necessary to model the data (to examine the longer time dependencies in the data).  Models with commensurate numbers of parameters performed similarly.  We will add further clarification of this to the text.
>
> **4. Missing some related work:**
>
> >I'd suggest the authors consider a conceptual comparison to DSARF [...]
>
> Thank you for the very relevant reference!  We have added qualitative comparison between SALT and DSARF to the revised manuscript.  We have also compared SALT to DSARF on the apnea example used by Farnoosh et al [2021].  This application is of particular interest because of the clinical relevance of the discrete states (respiration vs no respiration, see additional PDF).  We find that SALT performs comparably or slightly better than DSARF, achieving a normalized error of 22.57% vs DSARFs 23.86%, and producing qualitatively better discrete segmentations.  We will include these results and further discussion in a camera-ready version.
>
> ## R.E. Questions:
>
> **1. Selecting D using Prop. 1:**
>
> >How can we use Prop. 1 to decide which D value is optimal? [...]
>
> We explored using Proposition 1 to set D in Supplementary Figure 7, where we are able to predict the optimal rank for a SALT model when fitting to an LDS.  Proposition 1 further states that the accuracy (and consequently the marginal likelihood) is only dependent on the number of lags above a certain rank, a result which we verify in the additional PDF.  This result, in part, motivated the use of low-rank approximations.
>
> **2.  Hyperparameter fitting:**
>
> >Seems like SALT has more hyperparameters (H, L, and D) to select than either AR-HMM or SLDS [...]
>
> See General Response.  In general, we find SALT models faster and more robust to fit than SLDS models, and so even a three-way grid search takes a comparable amount of “user” and computational effort to the two-way SLDS search (see additional PDF).  In our experience, setting good hyperparameter ranges is straightforward with domain knowledge (eg. length of time dependencies).   We also believe that the simpler inference in SALT constitutes a “gain”, whereas the inference pipeline used in SLDS is a hidden hyperparameter which dramatically impacts performance.
>
> **3.  Runtimes:**
>
> >Can you report typical runtimes on the large datasets you study? [...]
>
> See General Response.  We have added typical runtimes for execution and training, and estimates for the time for hyperparameter tuning.  SALT models retain the runtime cost of ARHMMs, but the resistance to overfitting of SLDS, overall making hyperparameter selection easier.
>
> ## R.E. Limitations:
>
> **1. Discussion of limitations**:  See General Response.  We have added a more detailed limitations section touching on/ameliorating many of the points you raise.
>
> **Thank you again for your response.**  We ask if the reviewer would consider upgrading their score if we have successfully allayed your concerns.  Of course, if you have further questions, we are happy to answer them!
>
> --- The SALT authors.

---

> > ### Comment · Reviewer_6MVR · 2023-08-16
> > **Upgrading score to an accept**
> >
> > Thanks to the authors for the insightful response. I appreciate the artifacts in the PDF.
> >
> > Fig 1: The new comparison to DSARF, which suggests at least that
> > --- based on RMSE of reconstructions on one dataset, SALT is competitive with DSARF
> > --- the binary segmentations indeed do look better for SALT to my eye from a quick review
> >
> > Tab 1: The complete listing of hyperparameter settings for all experiments
> > --- This is helpful for reproducibility and ensuring fair comparisons across models. It relieves my original concerns about fairness.
> >
> > Tab 2: Runtimes
> > --- This helps clarify that ARHMM runtime at training is surprisingly high and that SLDS is definitely more costly than SALT (esp at inference)
> >
> > If the paper is revised suitably as promised (and includes these new artifacts), I am now persuaded to accept.
> >
> > ### Remaining questions
> >
> > Q0: Regarding L2-regularization of coefs for the AR-HMMs, can you confirm that the experiments in the paper did use this regularization?
> >
> > Your response seems to indicate this, but I can't find any clear description of your regularization strategy in the paper or the supplement.
> >
> > Please confirm. If you did use regularization already, I suggest revision to make clear how you pick the regularization strength, etc.
> >
> > Q1: Can you clarify what you mean by the following? Just that selecting between MCMC vs variational vs other methods is tough? Or that even within one choice, there are other "hidden" costs like learning rates?
> >
> > > the inference pipeline used in SLDS is a hidden hyperparameter which dramatically impacts performance.
> >
> > I think a clear description of the "hidden" costs of existing SLDS methods would help improve the paper

---

> > > ### Author Response · Authors · 2023-08-21
> > > **Clarifications**
> > >
> > > Excellent!  We are super happy to have allayed some of your concerns and answered some questions, and that you chose to raise your score.  In regard to your questions:
> > >
> > > Q0:  We did use L2 regularization in all ARHMM models in the paper. Thank you for the suggestion to include further details. In the updated paper, we will fully describe our investigation of regularization hyperparameters using a grid search.
> > >
> > > Q1:  Thank you for emphasising this, because it is an important plus-point for SALT that we under-discussed.  SLDS uses non-exact inference, such as MCMC or variational methods, where each method entails tuning parameters like learning rates, number of samples, optimizers etc, just to do inference in a model. So to concretely answer your question, even after selecting an inference method, a user still needs to select among many tuning parameters. In contrast, SALT uses simple and exact inference.  We will add explicit discussion of these hidden hyperparameters within SLDS inference approaches, why this is important, and how SALT is beneficial over SLDS approaches.
> > >
> > > Thank you again to the reviewer for their positive feedback and engagement!

---

### Official Review · Reviewer_sUaQ · 2023-07-06

**Soundness:** 3 good
**Presentation:** 3 good
**Contribution:** 3 good
**Rating:** 6
**Confidence:** 3

**Summary:**

This paper proposes a new time-series model called Switching Autoregressive Low-rank Tensor Model (SALT) that combines the advantages of autoregressive hidden Markov models and switching linear dynamical systems while addressing their weaknesses. SALT allows for longer range dependencies without overfitting and has been proven to be effective in various prediction tasks. The paper also explains the low-rank factorization used in SALT parameterization and provides experimental results demonstrating the effectiveness of SALT in various real-world applications.

**Strengths:**

1. Novelty: The paper proposes a new time-series model, SALT, that combines the advantages of two existing models while addressing their weaknesses. This is a novel approach that has not been explored before.

2. Clarity: The paper is well-written and easy to understand, even for readers who are not experts in the field. The authors provide clear explanations of the model and its components, as well as the experimental setup and results.

3. Empirical evaluation: The paper provides empirical evidence of the effectiveness of SALT in various real-world applications, including neural and behavioral time series. The authors compare SALT to other commonly used time-series models and demonstrate its superior performance.

4. Reproducibility: The authors provide code and data to facilitate reproducibility of their experiments. This is important for other researchers who want to build on this work or apply SALT to their own datasets.

5. Generalizability: The authors demonstrate the effectiveness of SALT in various real-world applications, suggesting that it is a generalizable model that can be applied to a wide range of time-series data.

**Weaknesses:**

1. Lack of theoretical analysis: While the paper provides empirical evidence of the effectiveness of SALT, it does not provide a detailed theoretical analysis of the model. This may limit the understanding of the model's properties and limitations.

2. Limited comparison to state-of-the-art models: While the paper compares SALT to other commonly used time-series models, it does not compare it to the most recent state-of-the-art models. This may limit the understanding of how SALT compares to the best-performing models in the field.

3. Limited discussion of limitations: The paper does not provide a detailed discussion of the limitations of SALT. This may limit the understanding of the situations in which SALT may not be the best choice for modeling time-series data.



**Questions:**

1. How generalizable are the results of this paper to other types of time-series data and applications?

2. How robust are the results of this paper to variations in the experimental setup, such as the choice of evaluation metrics or hyperparameters?

3. How does the complexity of SALT compare to other time-series models, and what are the implications of this for its practical use?

4. How does the choice of low-rank tensor regression in SALT affect its performance, and how might other regression methods be more appropriate for certain types of data?

5. What are the potential ethical implications of using SALT or other time-series models for analyzing sensitive or personal data, and how can these be addressed?

**Limitations:**

1. Limited scope: The paper focuses on a specific type of time-series data and does not explore the applicability of SALT to other types of time-series data. This may limit the generalizability of the model to other domains.

2. Limited sample size: While the paper provides empirical evidence of the effectiveness of SALT, the sample size of the experiments is relatively small. This may limit the generalizability of the results to larger datasets.

3. Limited evaluation metrics: The paper primarily evaluates the effectiveness of SALT using prediction accuracy metrics. While these are important metrics, they may not capture all aspects of the model's performance, such as its ability to capture complex temporal dependencies.

4. Limited discussion of hyperparameters: The paper does not provide a detailed discussion of the hyperparameters used in the experiments. This may limit the understanding of how sensitive the model's performance is to the choice of hyperparameters.

5. Limited discussion of computational complexity: While the paper briefly mentions the computational complexity of SALT, it does not provide a detailed analysis of the model's computational requirements. This may limit the understanding of the practical implications of using SALT in real-world scenarios with large datasets.

---

> ### Author Rebuttal · Authors · 2023-08-10
>
> We thank the reviewer for taking the time to read our submission and for their detailed and insightful feedback.  The five strength points were particularly heartening.  We now provide more detailed feedback.
>
> ## RE Weaknesses
>
> **1. Theoretical analysis:**
>
> >Lack of theoretical analysis: While the paper provides empirical evidence of the effectiveness of SALT, [...]
>
> We make a novel and broadly applicable link between low-rank tensor regressions and linear dynamical systems in Proposition 1, linking the core components of SALT and SLDS models.  Furthermore, SALT models are still fundamentally ARHMMs, allowing us to directly use existing results and methods to further understand and control the behavior of SALT models, eg, enforcing sparsity [Shah et al, 2015, NeurIPS] and non-negativity of the factors [Shashua and Hazan, 2005, ICML].
>
> **2. Choice of baselines:**
>
> >Limited comparison to state-of-the-art models: While the paper compares SALT [...]
>
> See General Response B.  We compare SALT to SLDS, ARHMMs and TVART (in the supplement).  As raised by 6MVR, we also compared to DSARF on an example from the original paper, and find that SALT performs comparably if not better than this deep method.
>
> We note that the relevant comparisons are models that yield a hybrid continuous-discrete representation.  Many common methods are therefore not directly applicable.  If the reviewer has suggestions of models with this structure, we are happy to include a comparison ahead of any camera ready version.
>
> **3. Discussion of Limitations:**
>
> >Limited discussion of limitations: The paper does not provide a detailed discussion of the limitations of SALT. [...]
>
> See General Response (A).  We have included thorough discussion of SALTs limitations.
>
> ## RE Questions
>
> **1. Generalizability of SALT:**
>
> >How generalizable are the results of this paper to other types of time-series data and applications?
>
> We believe the analysis allowed by SALT is fully generalizable, particularly when the interdependencies between observations are insightful to model and a mixed discrete-continuous representation is sought.  The ease of tuning and fitting SALT models make it an excellent option for exploratory data analysis.  Beyond the two time-series applications in the paper, we added a third on Sleep Apnea data experiment from DSARF (as pointed to by 6MVR, see additional PDF), further demonstrating the generalizability of SALT.
>
> **2. Robustness to hyperparameters and metrics:**
>
> >How robust are the results of this paper to variations in the experimental setup, such as the choice of evaluation metrics or hyperparameters?
>
> See General Response.  SALT models are fairly robust to the choice of hyperparameters (within reason).  There are also fewer hyperparameters, as there is no learning rate or decay schedule required, no tuning of variational/interactive/approximate inference algorithms, and EM guarantees convergence to a (local) maximum so there is no need for tuning early stopping.
>
> With regard to metrics, we considered multiple different metrics for each model.  We found that the use of confusion matrices, segmentations and marginal log-likelihoods (where applicable) yielded comparable results across methods and applications.
>
> **3. Complexity of SALT:**
>
> >How does the complexity of SALT compare to other time-series models, and what are the implications of this for its practical use?
>
> The complexity of SALT lies between the ARHMM and the SLDS.  The ARHMM is difficult — often intractable —  to fit because of overparameterization.  The SLDS is tricky to fit because it requires iterative variational inference methods.  SALT admits fast computation of an exact marginal likelihood, allowing fair comparison **between** different model classes.   In contrast, SLDS estimates a bound and is dependent on approximate or variational inference schemes.
>
> SALT is also easier to implement and more extensible than SLDS.  Each new SLDS model variant requires deriving and tuning a new variational inference scheme.  New SALT models can be derived by simply updating the update equations (falling back to coordinate gradient descent).  As in Weakness 1, SALT can therefore directly leverage the rich literature in tensor factorization methods.
>
> **4. Role of rank in optimization:**
>
> >How does the choice of low-rank tensor regression in SALT affect its performance [...]
>
> The rank trades off expressivity with overfitting, acting as an implicit regularizer, in contrast to other models that require additional regularizers and hyperparameters to prevent overfitting.  As noted in General Response B, there is no consensus on the best way to regularize an ARHMM, so SALT actually provides an intuitive regularization.
>
> **5. Ethical concerns:**
>
> >What are the potential ethical implications of using SALT [...]
>
> We do not believe there are any new ethical implications from SALT.
>
>
> ## RE Limitations
>
> 1. **Type of evaluation**: See Questions 1.
>
> 2. **Scope of evaluation**: We tested SALT on a number of standard benchmarks common to this domain.  We have also tested SALT on the apnea dataset used in DSARF (noted by Reviewer 6MVR).  SALT outperforms DSARF in both reconstruction accuracy and segmentation quality.
>
> 3. **Temporal dependencies**: Quantifying temporal dependencies is tricky.  We explored this in the C. elegans example, using a known neural function as a proxy for capturing time dependencies.  We have added some clarification on this to the results section.
>
> 4. **Hyperparameters**: See General Response / Questions 2.
>
> 5. **Computational complexity**: See General Response and additional PDF.  SALT models are broadly the same speed or faster to train and apply than ARHMMs and SLDSs.
>
> **Thank you again for your response.**  If we have successfully allayed your concerns, we ask if you would consider upgrading your score.  Of course, if you have further questions, we are happy to answer them!
>
> --- SALT authors.

---

### Official Review · Reviewer_BTJL · 2023-07-07

**Soundness:** 2 fair
**Presentation:** 2 fair
**Contribution:** 2 fair
**Rating:** 4
**Confidence:** 4

**Summary:**

The paper proposes Switching Autoregressive Low-Rank Tensor (SALT) models, a variant of an Autoregressive Hidden Markov Model (ARHMM) in which the model’s temporal dynamics are captured by a low-rank tensor approximation, thus combining the parameter efficiency of Switching Linear Dynamical Systems (SLDS) with the simple learning and inference techniques available for ARHMMs. The paper further shows that a SALT model can approximate a stable LDS model to a degree that depends only on the eigenspectrum of the LDS model and the order of the SALT model.


**Strengths:**

- The paper makes two main contributions: (1) a parametrization of ARHMM dynamics with a low-rank Tucker decomposition; and (2) a theoretical analysis of the resulting model’s approximation error relative to a stable LDS. The latter is an interesting theoretical insight and includes a rigorous proof in Appendix B.

- The background section (Section 2) provides a useful recap of (switching) autoregressive models and (switching) linear dynamical systems that motivates the need for parameter-efficient architectures with tractable learning and inference algorithms.

- The experiments (Section 5) confirm that SALT models can learn (S)LDS dynamics and require less training data than their ARHMM parent due their parameter-efficient representation. The experiments on real-world data demonstrate that SALT models can learn semantically meaningful filters / state representations and outperform ARHMMs in terms of test log-likelihood.

**Weaknesses:**

- As discussed in Section 4, structural decompositions of temporal dynamics, including low-rank approximations, have been explored in a variety of different contexts. This includes the representation of the autoregressive tensor as a Tucker decomposition and, while I appreciate the differences in the graphical structure compared to these models and the theoretical insights provided in Proposition 1, the technical contribution of this paper is not particularly strong.

- Although the core ideas of this paper are relatively simple, the confusing tensor notation makes the paper unnecessarily hard to follow. Uncommon operations like the $n$-mode tensor-matrix product or the $n$-mode tensor matricization must be properly defined, ideally including visual illustrations. Having to figure out which high-dimensional slices are being multiplied is a burden on the reader and diverts the focus from the underlying ideas. Where possible, I would recommend to express the model dynamics in summation notation instead of tensor operations; it would greatly improve readability.

- Ultimately, the representation of the autoregressive tensor with a Tucker decomposition is a structural assumption that modulates the spectrum between flexible models prone to overfitting and robust models prone to large bias. Since SALT models cannot be more expressive than a generic ARHMM, I view them primarily as a form of implicit regularization and, as such, would have liked to see a comparison with other regularization techniques (e.g., a Bayesian treatment with strong priors).

- The experimental validation compares the proposed method to two traditional time-series models (ARHMM and SLDS) but does not include any state-of-the-art baselines (e.g., based on deep variants of autoregressive or state-space models, Gaussian processes, Transformer architectures, normalizing flows, etc.). Even if the proposed method is more related to ARHMM or SLDS, it is expected that the evaluation takes other time-series architectures into account, including more competitive and more recent developments. The data is relatively simple as well, with two toy datasets consisting of (S)LDS simulations and a small number of real-world sequences (3 mice, 1 worm).

**Questions:**

Minor comments:

- It would improve the accessibility of the paper if it included graphical models of (S)LDS and (S)VAR, similar to Figure 1, so that the advantages and disadvantages of the different model classes can be analyzed in terms of their (marginalized) conditional independence assumptions.

- I wish the presentation had pointed out the strong connections of the Tucker decomposition to (higher-order) PCA/SVD. In contrast to the Tucker decomposition, this is something most readers are familiar with and would have helped with the intuition of an orthogonal/unitary factor approximation.

Questions:

- Section 5.1: how were the random matrices sampled?
- Figure 2(A): why does a rank *higher* than 7 (Tucker-SALT) and 10 (CP-SALT) lead to *worse* approximations. Should the MSE not be strictly decreasing?

**Limitations:**

- The paper briefly mentions possible extensions but does not include an explicit discussion of limitations.

- The paper does not discuss any ethical concerns related to the proposed method.

---

> ### Author Rebuttal · Authors · 2023-08-10
>
> # Response to Reviewer BTJL
>
> We thank the reviewer for taking the time to read our submission and for their detailed and insightful feedback.  The strengths you outline really neatly encapsulated our objectives, and so that was great to hear!  We will now provide some more detailed feedback beyond the General Response provided above.
>
> ## R.E. Weaknesses:
>
> **1.  Technical Contributions:**
>
> >[...]  While I appreciate the differences in the graphical structure compared to these models and the theoretical insights provided in Proposition 1, the technical contribution of this paper is not particularly strong.
>
> While the technical contributions of our work may not be a paradigm shift, we are confident it will be useful to many practitioners who regularly use these types of models.  SALT sits at the intersection of several widely-used methods, leveraging efficient components and capturing the benefits of each model to yield a model that offers a unique insight into the system.  We also release our fast JAX code.  We also highlight, as raised in General Response B, that inference in SLDSs is not a “solved” problem and often requires complex, hard-to-tune inference schemes – see recent publications such as Zoltowski et al [2020, ICML] and Berger et al [2022, NeurIPS].  Therefore, achieving a parameter complexity comparable with SLDS, while retaining fast and exact inference, and bringing additional benefits, is a valuable contribution to the field that will be of particular interest to the NeurIPS readership.
>
> **2.  Improving Notation**
>
> >Although the core ideas of this paper are relatively simple, the confusing tensor notation makes the paper unnecessarily hard to follow. [...]
> See General Response A.  We have simplified and clarified the notation where possible, and added diagrammatic explanations to the supplement.  Thank you for this feedback.
>
> **3.  SALT as implicit regularization:**
>
> >Ultimately, the representation of the autoregressive tensor with a Tucker decomposition is a structural assumption that modulates the spectrum between flexible models prone to overfitting and robust models prone to large bias. [...]
>
> All models trained used L2 regularization, including the ARHMM (we have added details to the supplement).  Further to our response in General Response B, we note that there is no clear consensus on the correct way to regularize the parameters of vector autoregressive hidden Markov models – especially ones with low-rank factors.  Therefore, SALT provides a natural and intuitive way to control the expressivity of the model family.  Nonetheless, exploring more powerful Bayesian regularization or hierarchical components to SALT are important directions for future research.
>
> **4.  Empirical evaluation:**
>
> >The experimental validation compares the proposed method to two traditional time-series models (ARHMM and SLDS) but does not include any state-of-the-art baselines (e.g., based on deep variants of autoregressive or state-space models, Gaussian processes, Transformer architectures, normalizing flows, etc.). [...]
>
> See General Response B.  We stress, our objective with this work was not to create a state-of-the-art regression model.  Instead, SALT models provide an “interpolation” between the widely used SLDS and ARHMM.  These models provide a very experimentally valuable hybrid continuous-discrete representation, and are used regularly by practitioners owing to their simplicity, efficiency and utility.  As such, SALT is a valuable model that allows practitioners to retain the benefits of these models, while flexibly ameliorating their individual weaknesses.
>
> The tasks we examined were taken from the literature, but, if the reviewer has suggestions for additional experiments then we are more than happy to add them ahead of a camera-ready version.  Since receiving the reviews, we have compared SALT to DSARF (a “deep” switching model, as suggested by 6MVR) on the apnea example from that paper.  SALT outperforms comparably than DSARF in terms of reconstruction accuracy and segmentation quality.  These results are included in the additional PDF.
>
> ## Minor Comments:
>
> 1.  **Visualization of model families**:  We have added extra graphical models for each model to the supplement, and discussed the implications in the main text.  Thank you for the suggestions.
>
> 2.  **Links to SVD**:  This is a great connection, we have added discussion of the links to SVD to the background, and also re-highlighted them when we discuss the factors later on.
>
> ## Questions:
>
> 1.  **How were random matrices sampled**:  Matrices were sampled as random rotational matrices.
>
> 2.  **Why can increasing rank reduce performance**:  Great observation – the answer is actually overfitting to the training data.  Here we are plotting the error on held-out test data.  The degradation also exists for the log-likelihood but is less visible.  We have added clarification on this.  We stress that this overfitting is caught by cross-validation.
>
> ## Limitations:
>
> 1.  **Inclusion of limitations**:  See General Response A.  We have added discussion of limitations including model mismatch and missing observations.
>
> 2.  **Inclusion of ethical concerns**:  We have added clarification that we see no ethical concerns specifically related to SALT.
>
>
> **Thank you again for your response.**  If we have successfully allayed your concerns, we ask if you would consider upgrading your score.  Of course, if you have further questions, we are happy to answer them!
>
> --- The SALT authors.

---

> > ### Comment · Reviewer_BTJL · 2023-08-14
> >
> > I want to thank the authors for their insightful response. While it does address some of my concerns (e.g., notation, experiment details), I still feel the paper’s contribution and evaluation are relatively weak. At its core SALT is a specific type of ARHMM parametrization that allows for efficient learning/inference, with the rank controlling the model’s expressivity. I appreciate the new comparison to DSARF, but I do think the paper requires comparisons to other types of structural constraints (e.g., priors, loss penalties, hand-crafted dynamics).

---

> > > ### Author Response · Authors · 2023-08-15
> > >
> > > We thank the reviewer for the response. We are glad that our previous response was able to address some of your concerns.
> > >
> > > We would like to re-emphasize that the paper's contribution is not simply a tensor-factorized ARHMM, but also the relationship between this model and widely used SLDS models. We make a novel and broadly applicable link between low-rank tensor regressions and linear dynamical systems in Proposition 1, linking the core components of SALT and SLDS models. Elucidating this relationship and demonstrating the advantages of fitting SALT models over SLDS is an important contribution to the field that will be of particular interest to the NeurIPS readership.
> > >
> > > With regard to other types of structural constraints for vector autoregressive (hidden Markov) models, aside from L2 regularization on the parameters, there is no clear precedent for other types of regularization (See General Response B). Many regularizers and priors are difficult to work with, and are not widely used in practice. Moreover, we are not claiming that one couldn't get similar performance through ARHMMs with other types of structural constraints. We are instead exploring tensor factorization as a type of constraint, and in doing so, we not only identified an effective model class, but also bridged the gap between the widely used SLDS and ARHMM.
> > >
> > > **Again, thank you for your feedback**. If there are any further questions, please do not hesitate to ask.
> > >
> > > --- The SALT authors

---

### Author Rebuttal · Authors · 2023-08-10

Thank you to all four reviewers for taking the time to read our submission and provide insightful and constructive feedback. We presented Switching Autoregressive Low-Rank Tensor (SALT) Models, which combine the benefits of ARHMMs and SLDS models (such as parameter efficiency, fast exact inference and interpretability), while ameliorating the drawbacks of each method individually. We compared SALT to similar models across a range of problems.

Here we respond to two themes that were touched on by multiple reviews: clarity and experimental utility. We then respond to individual reviews in more detail below each review.

## (A) Clarity

Reviewers 4hZ1 and sUaQ commended the clarity of our submission. However, there were comments on the complexity of the notation, the explanation of hyperparameter tuning, and the discussion of limitations.

**R.E. Notation:** We generally followed the tensor notation of Kolda and Bader [17]. However, we agree that tensor notation can sometimes be a lot to process. We have simplified the notation and used summation operators in several places. We have also added explicit definitions of tensor operations (with diagrams!).

**R.E. Hyperparameter tuning:** We conducted extensive hyperparameter tuning for all models using grid searches, and have included extensive details. Generally, SALT models are fairly robust to hyperparameter settings and across seeds. ARHMMs are very robust across seeds for good hyperparameters, but more are more susceptible to bad hyperparameter settings (resulting in very poor and more variable performance). The best SALT models use similar dimensions commensurate to the best SLDS models. We include exact dimensions in the additional PDF.

We also note that the number of hyperparameters in SALT is actually notably lower than many modern “deep” or kernel-based methods. Closed form updates alleviate the need for tuning learning rates; there are only three architectural dimensions that need tuning (compared to many layers widths, activations etc for deep methods); and exact inference removes any hyperparameters required by approximate or variational inference methodologies. Therefore, we assert that SALT actually has *fewer* parameters to tune than comparable models.

**R.E. Limitations:** We have added extensive discussion of the limitations of SALT, including:
- Model mismatch: SALT is a simpler model family compared to (eg.) DSARF. This simplicity accelerates optimization and reduces overfitting, but may increase the bias in predictions if the SALT model used is poorly tuned.
- Missing observations: SALT cannot natively handle missing observations. We considered using linear interpolation to bootstrap SALT, but believe that a more principled method for handling missing observations should be possible because of the nature of SALT models.
- Bayesian treatment: It is not currently possible to share information between time series with different dimensional observations.  “Hierarchical SALT” is a possible extension to tackle this.

**R.E. Ethics:** We have added a short discussion outlining that there are no new ethical concerns as a result of SALT.

## (B) Experimental Utility

Several reviewers commented on the baselines we compared to. We note that a hybrid continuous-discrete representation of the data is experimentally valuable. For instance, MoSeq [Wiltschko et al, 2015] uses ARHMMs to segment mouse behavior into discrete labels and a continuous state. The requirement precludes many common models (transformers, GPs, RKNs etc) which lack a discrete component. Our objective was not to create a state-of-the-art regression model, but rather explore the space between two widely used models that offer this description.

We highlight that inference in SLDSs is not a “solved” problem and an active area of research. Numerous inference approaches exist, often requiring approximate or hard-to-tune inference schemes. Recent examples include Laplace-EM [Zoltowski et al, 2020, ICML] and linear programming [Berger et al, 2022, NeurIPS]. Developing new SLDS model variants therefore requires a deep understanding of the associated inference techniques. In contrast, new SALT models can be derived by simply updating the update equations (which can always fall back to coordinate gradient descent if a closed-form update cannot be derived).  SALT can therefore tap easily into the rich existing literature on tensor regressions, such as enforcing sparsity [Shah et al, 2015, NeurIPS] or non-negativity [Shashua and Hazan, 2005, ICML], in a way that SLDS cannot.

Similarly, to our knowledge, there is no clear consensus on the best way to regularize vector autoregressive (hidden Markov) models; several possibilities exist, see, eg, Melnyk and Banerjee [2016, ICML] or Ni and Sun [2005, ASA]. Many regularizers and priors are difficult to work with, and are not widely used in practice. Beyond this, even well-regularized ARHMMs do not natively capture interpretable low-dimensional dynamics, as both SALT and SLDS models do (see Figure 3). These low-dimensional dynamics are experimentally as useful as the discrete segmentation.

As also highlighted above, SALT has fewer hyperparameters than other methods, when methods are considered in their entirety. Therefore, SALT models combine the benefits and ameliorate the weaknesses of both ARHMMs and SLDSs, and provide an accessible, extensible, interpretable and performant alternative that can be easily tuned and deployed by practitioners.

Finally, we have also compared SALT to DSARF (suggested by 6MVR) on the apnea task included in Farnoosh et al [2021] and have added these results and discussion. SALT matches or exceeds the performance of DSARF in terms of the NRMSE% on held-out test data and the quality of the segmentation (see attached PDF).

**Again, thank you for taking the time to review our paper**. If there are any further questions, please do not hesitate to ask!

--- The SALT authors

---

### Decision · Program_Chairs · 2023-09-21

**Decision:**

Accept (poster)

**Comment:**

This paper describes an extension to switched linear dynamical systems where rather than a 1-step AR model, the switched system is an N-tap model. The method, named SALT, switches a low-rank tensor representing the system. The method is leverages low CP- or Tucker-rank representations of the AR-P model, and provides efficient fitting procedures as well as comparisons to toy data (similar to prior work in SLDS), mouse behavior data, and C. elegans data.

Overall the reviewers were generally positive about the contributions. Reviewer BTJL however brings up the point that the general model is fairly incremental to past work in ARHMMs, and questioned if the modeling and supporting experiments support the method as a valuable contribution to the conference. The reviewer does make good points, especially that the main benefit in some of the experiments seems to primarily be in increased modeling efficiency rather than new insights. The authors responded by noting that often modeling choices, such as the ability to compare Tucker and CP with other ARHMM assumptions, as well as the release of efficient code for the community do serve as a contribution even if the model is not completely new. Given the discussion and especially the support of the other reviewers, I recommend this work for publication.